# Population Structure and Phylogeography of Marine Gastropods *Monodonta labio* and *M. confusa* (Trochidae) along the Northwestern Pacific Coast

**Yuh-Wen Chiu** [1,*], **Hor Bor** [2], **Jin-Xian Wu** [3], **Bao-Sen Shieh** [4] and **Hung-Du Lin** [5]

1 Department of Biological Resources, National Chiayi University, Chiayi 600, Taiwan
2 Environmental Protection Department, New Taipei City Government, New Taipei 220, Taiwan; emilebor@gmail.com
3 Guangzhou Key Laboratory of Subtropical Biodiversity and Biomonitoring, School of Life Science, South China Normal University, Guangzhou 510631, China; jin1790274@163.com
4 Department of Biomedical Science and Environmental Biology, Kaohsiung Medical University, Kaohsiung 807, Taiwan; bsshieh@kmu.edu.tw
5 The Affiliated School of National Tainan First Senior High School, Tainan 701, Taiwan; varicorhinus@hotmail.com
* Correspondence: chiuywlab@gmail.com; Tel.: +886-5-2717760

**Abstract:** The genetic structure and demographic history of marine organisms are influenced by biological and ecological features, oceanic currents, and the paleo-geological effects of sea-level fluctuations. In this study, we used mitochondrial COI + 16S gene analysis to investigate the phylogeographic pattern and demography of *Monodonta labio* and *M. confusa* in Taiwan, the Ryukyu Islands, Japan, mainland China, and Korea. Our genetic analysis identified two major lineages that were not evident from the distribution patterns of different populations. The Taiwan Strait, which acted as a land bridge during Pleistocene glaciations, and the lack of strong dispersal barriers to gene flow between ocean basins after glaciations shaped the phylogeographic pattern. The genetic differentiation in the Ryukyu Islands was influenced by the specialist-generalist variation hypothesis and the Kuroshio Current. Bayesian skyline plot analyses suggested that the effective population size of *M. labio* and *M. confusa* rapidly increased approximately 0.1 and 0.075 million years ago, respectively. Our approximate Bayesian computation analysis suggested that all *M. labio* and *M. confusa* populations experienced a decline in population size following a recent population expansion and constant size, respectively. Our study provides a baseline for future investigations of the biogeographical patterns of marine gastropods in the Northwest Pacific and offers valuable insights for the management, sustainable resource utilization, and conservation of this species.

**Keywords:** DIY-ABC; mitochondrial; *Monodonta labio*; phylogeography

## 1. Introduction

Global climate conditions have changed during glacial-to-interglacial periods, leading to the oscillation of the sea level, oceanic current pattern, and life history traits (e.g., dispersal ability and life cycle), which could have an unavoidable effect on the phylogeography, demographic history, and population genetic diversity of coastal marine systems [1,2]. The Northwest Pacific, mainly including the East China Sea (ECS), the South China Sea (SCS), the Yellow Sea, and the Sea of Japan, is an epicenter of marine biodiversity and provides an excellent model system to investigate as a major glacial refuge, driving population differentiation at different spatial and temporal scales in some marine species during glacial periods of the Plio-Pleistocene [3]. During major glaciations, when there is a sea level decline of approximately 120–140 m below the present level, the northwestern (NW) Pacific is thought to form a continuous land mass, effectively closing the sea passage (the Korean Strait and Taiwan Strait). Because of their geographical isolation, the East China Sea (ECS) and the

South China Sea (SCS) are partially enclosed, with connections to the Pacific occurring through the Okinawa Trough and the Bashi Strait in a restricted manner. The emergence of the Strait as a barrier is one of the significant factors that have shaped the current phylogeographical patterns and the presence of deeply divergent lineages in marine species, such as marine gastropods (*Reticunassa festiva*) [4], grey mullet (*Mugil cephalus*) [5], spotted scat (*Scatophagus argus*) [6], and cutlassfish (*Lepturacanthus savala*) [7]. The genetic structure of numerous marine organisms may be influenced by larval dispersal, which is believed to be affected by the ocean current system in the northwest Pacific. In general, ocean currents can facilitate the migration of marine organisms over long distances and enhance population connectivity. Conversely, convergent ocean currents can create an effective barrier to gene flow to a certain degree [8]. Prominent ocean currents in the NWP and its marginal seas include the cyclonic circulation of the Kuroshio Current (KC) and coastal currents. The Kuroshio Current is characterized by a warm current that originates from the Philippines and flows northeastward to the Pacific coast of southern mainland Japan via Taiwan and the Ryukyu Islands. The NWP marginal seas are characterized by complex coastal currents, such as the China Coastal Current (CCC), Tsushima Current (TC), and Korean Coastal Current (KCC). By altering the dynamics of larval supply, enhancing population connectivity, or producing different outcomes from the sea surface temperature gradient, ocean currents play an important role in shaping genetic structure. Meanwhile, some phylogeographic studies in marine organisms have shown a lack of genetic structuring among distant populations with continuous gene flow via current-driven larval dispersal, such as in bigeye tuna (*Thunnus obesus*) [9], Chinese beard eel (*Cirrhimuraena chinensis*) [10], and the large yellow croaker (*Larimichthys crocea*) [11].

The marine trochid gastropod *Monodonta* belongs to the family Trochidae and is commonly found in the mid-intertidal zone. It exhibits a wide distribution in the Pacific Northwest. This herbivorous snail is known to feed on microalgae [12]. The genus *Monodonta* consists of eleven species and one subspecies, of which eight species and one subspecies are distributed in East Asia: *M. australis*, *M. canalifera*, *M. confusa*, *M. glabrata*, *M. labio*, *M. viridis*, *M. neritoides*, *M. perplexa perplexa*, and *M. perplexa boninensis* (listed in Donald [12]), including two species (*M. confusa* and *M. perplexa*) that undergo a planktonic stage lasting 3 days in their life cycle [13]. Biological traits such as effective dispersal capability through a planktonic larval stage and life cycle can strongly influence the genetic differentiation and speciation of marine organisms [14]. In general, the presence of a long-lived planktonic stage is often considered important for dispersal and lower geographic variations, such as *Sicyopterus japonicus* [15] and *Thais clavigera* [16]. In contrast, a short planktonic stage in marine organisms has revealed a clear genetic structure, such as in the Japanese turban shell (*Turbo (Batillus) cornutus*) [17] and the Moon Turban Snail (*Lunella granulate*) [18]. Previous studies of *M. labio* detected intraspecific variations in the shell morphology and genetic structure in NWP [19]. According to Zhao et al. [20], *M. labio* can be divided into five primary clades distributed along the Chinese coastline, and the Dongshan land bridge served as a separation between the ancient East China Sea and the ancient South China Sea, forming two possible refugia. Another similar study proposed that *M. labio* and *M. confusa* were primarily separated into different lineages with a boundary between the Japanese mainland and the Ryukyu Islands [21]. The genetic structure of *M. labio* and *M. confusa* is also influenced by ecological factors such as habitat specificity and adaptability. Molecular studies have indicated that *M. labio* displays lower genetic diversity and higher genetic differentiation compared to *M. confusa*, potentially influenced by specific ecological factors [22]. This pattern is known as the "specialist-generalist variation hypothesis" (SGVH) [22]. *Monodonta labio* and *M. confusa* have been regarded as excellent model systems in the NWP for examining the impact of geographical and ecological isolation on genetic divergence. Previous studies of genetic structure in *M. labio* and *M. confusa* focused on mainland China, the Ryukyu Islands, and Japan, but provided little information on Taiwan. Furthermore, the phylogenetic processes of *M. labio* and *M. confusa* on mainland China, Taiwan, the Ryukyu Islands, and Japan have not been

thoroughly or convincingly elucidated, and there have been no investigations into the population relationships between Taiwan and other regions. Statistical phylogeography is a powerful approach to understanding the processes that have shaped the spatio-temporal demographic and distributional changes in *M. labio* and *M. confusa*. Relying on the former molecular results, our objective was to use ABC simulations, which offer a framework for testing competing hypotheses to determine the best model of population divergence and demographics that fits the patterns of key demographic parameters, such as changes in effective population size through time. This study aimed to compare the genetic variation between two co-occurring species of intertidal snails, *M. labio* and *M. confusa*, from the genus Monodonta, found on the Northwestern Pacific Coast. To achieve this, we utilized the mitochondrial COI + 16S genes to establish the phylogeographic patterns of these species across mainland China, Taiwan, the Ryukyu Islands, and Japan. The three main objectives were to: (1) determine the genetic diversity and structure of *M. labio* and *M. confusa*; (2) reconstruct the phylogeographic patterns of *M. labio* and *M. confusa*, understand the influence of paleoclimate oscillations on their current distribution; and (3) examine the demographic response of *M. labio* and *M. confusa* to repeated climate changes during the Quaternary period.

## 2. Materials and Methods

### 2.1. Sample Collection and Sequencing

The mitochondrial COI + 16S gene, which was 1340 bp long, was sequenced from 85 specimens of *M. labio* and 41 specimens of *M. confusa* collected from thirteen localities in Taiwan, mainland China, Jeju Island in Korea [was missing], and representing the Ryukyu Islands and more northern localities within Japan (Table 1 and Figure 1). Based on their geographical distribution, the thirteen populations were classified into four regions: Taiwan Island (Audi, AD; Miaoli, WW; Tainan, CG; Taitung, TD); Ryukyu Islands (Okinawa, OK; Ishigaki, SY); Japan (Inubosaki, CF; Ine, YG; Kagoshima, SI); Mainland China (Kinmen, KM; Matsu, MZ); and Korea (Jeju, GJ) (Table 1 and Figure 1). Table 1 provides details on the location and sample sizes.

**Table 1.** List of sampling locations and the sample sizes, the number of haplotypes (N), haplotype diversities (h), and nucleotide diversities of the mtDNA COI + 16S region sequences for each location.

| Locations | Sampling Location | Code | Coordinates | *N* | *Nh* | *h* | $\theta\pi$ | $\theta\omega$ |
|---|---|---|---|---|---|---|---|---|
| | *Monodonta labio* | | | 85 | 61 | 0.987 | 0.0131 | 0.0151 |
| Taiwan Island | | | | 22 | 20 | 0.991 | 0.0.0448 | 0.0376 |
| | Audi | AD | 121.92 25.08 | 5 | 5 | 1.000 | 0.0032 | 0.0035 |
| | Miaoli | WW | 120.73 24.61 | 6 | 6 | 1.000 | 0.0041 | 0.0052 |
| | Tainan | CG | 120.10 23.12 | 11 | 10 | 0.982 | 0.0019 | 0.0028 |
| Penghu | | | | 9 | 7 | 0.917 | 0.0017 | 0.0027 |
| | Penghu | PH | 119.58 23.58 | 9 | 7 | 0.917 | 0.0017 | 0.0027 |
| Coast of mainland China | | | | 15 | 12 | 0.962 | 0.0179 | 0.0387 |
| | Kinmen | KM | 118.35 24.43 | 8 | 7 | 0.964 | 0.0315 | 0.0480 |
| | Matsu | MZ | 119.98 26.22 | 7 | 7 | 1.000 | 0.0027 | 0.0030 |
| Japan | | | | 15 | 10 | 0.895 | 0.0015 | 0.0029 |
| | Ine | YG | 135.29 35.70 | 9 | 7 | 0.917 | 0.0012 | 0.0019 |
| | Kagoshima | SI | 130.46 32.51 | 6 | 5 | 0.933 | 0.0018 | 0.0022 |
| Ryukyu | | | | 24 | 17 | 0.964 | 0.0037 | 0.0056 |
| | Okinawa | OK | 127.79 24.25 | 10 | 9 | 0.978 | 0.0035 | 0.0044 |
| | Ishigaki | SY | 124.11 24.36 | 14 | 8 | 0.901 | 0.0032 | 0.0035 |
| | *Monodonta confusa* | | | 41 | 33 | 0.983 | 0.0082 | 0.0115 |
| Taiwan Island | | | | 12 | 12 | 1.000 | 0.0.0086 | 0.0096 |
| | Audi | AD | 121.92 25.08 | 7 | 7 | 1.000 | 0.0093 | 0.0097 |
| | Taitung | TD | 121.19 22.82 | 5 | 5 | 1.000 | 0.0089 | 0.0086 |

**Table 1.** *Cont.*

| Locations | Sampling Location | Code | Coordinates | *N* | *Nh* | *h* | θπ | θω |
|---|---|---|---|---|---|---|---|---|
| Penghu | | | | 5 | 5 | 1.000 | 0.0093 | 0.0093 |
| | Penghu | PH | 119.58 23.58 | 5 | 5 | 1.000 | 0.0093 | 0.0093 |
| Coast of mainland China | | | | 7 | 6 | 0.952 | 0.0050 | 0.0067 |
| | Matsu | MZ | 119.98 26.22 | 7 | 6 | 0.952 | 0.0050 | 0.0067 |
| Japan | | | | 13 | 10 | 0.923 | 0.0037 | 0.0050 |
| | Inubosaki | CF | 140.87 25.70 | 10 | 7 | 0.867 | 0.0031 | 0.0034 |
| | Ine | YG | 135.29 35.70 | 3 | 3 | 1.000 | 0.0049 | 0.0049 |
| Korea | | GY | | 4 | 4 | 1.000 | 0.0042 | 0.0044 |
| Total | | | | 126 | 94 | 0.992 | 0.0825 | 0.0441 |

N: sample size π: nucleotide diversity, Nh: number of haplotypes h: haplotype diversity.

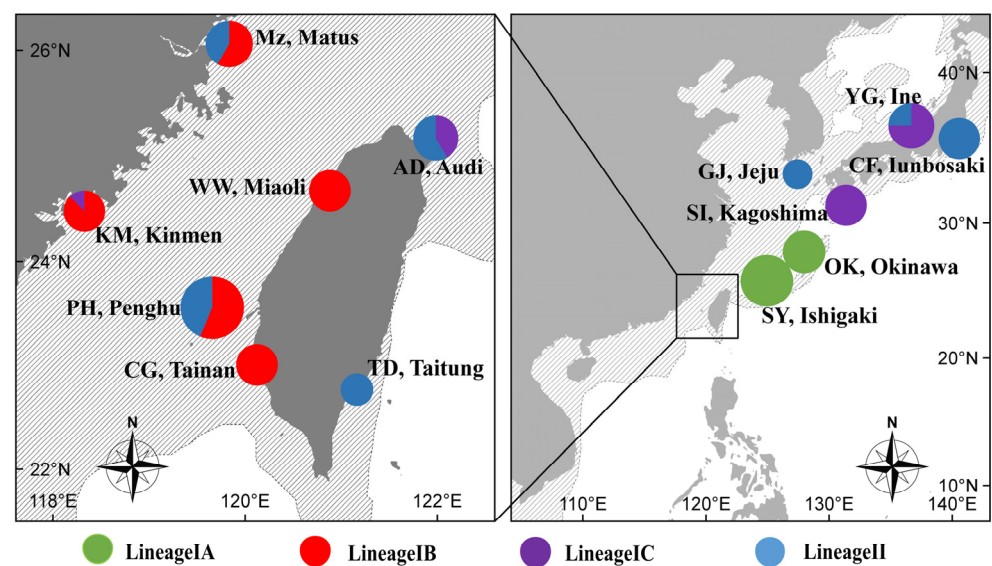

**Figure 1.** Map showing the 13 sampling localities of *Monodonta labio* (Lineage I) and *M. confusa* (Lineage II) examined in this study. Collection sites (circule) correspond to locations given in the text and Table 1. The frequencies of the haplogroups (Figure 3) in each population are displayed on the map. The areas shaded with diagonal lines in the figure represent the regions that were exposed at sea level during the glacial period, indicating the coastline at that time.

Samples were manually collected from the field sites, fixed, and stored in 95% ethanol. All animal experiments were conducted in the laboratory of Yuh-Wen Chiu at National Chiayi University, following the guidelines and approval of the Animal Research and Ethics Committee of the same university. The use of gastropods as experimental animals in Taiwan does not require a license. Muscle tissue was used to extract genomic DNA using a tissue and cell genomic DNA purification kit ((GeneMark Technology Co., Tainan, Taiwan)). Fragments of the mitochondrial cytochrome subunit I gene (COI) and 16S rRNA region (16S) were amplified by polymerase chain reaction (PCR) using the primers LCO-1490 (5′- GGTCAACAAATCATAAAGATATTGG-3′) and HCO-2198 (5′-TAAACTTAGGGTGACCAAAAATCA-3′) [23] for COI and 16Sar (5′-CGCCTGTTTATCA AAAACAT-3′) and 16Sbr (5′-CCGGTCTGAACTCAGATCACGT-3′) [24] for 16S. The polymerase chain reaction (PCR) was performed in a 25 μL reaction volume according to the conditions specified in Chiu et al. (2013) [18]. Subsequently, the PCR products were run on an ABI 377 automated sequencer (Applied Biosystems, Foster City, CA, USA). All sequences have been deposited in GenBank under the following accession numbers: OQ880826-OQ 880919. Other sequences of *M. labio* and *M. confusa* are available on Genbank [20–22], with additional sequences of *M. perplexa* selected as the outgroup.

### 2.2. Sequence Alignment and Data Analysis

We aligned the COI + 16S genes of the mtDNA sequences using CLUSTAL X v. 2.0 software [25] and manually adjusted the alignment. To calculate the genetic diversity of each population, we used DnaSP v5.0 software [26] and determined the number of haplotypes (N), haplotype diversity (h), private haplotypes (ph), nucleotide diversity ($\pi$), and nucleotide diversity ($\theta$) [27]. To examine the spatial partitioning of genetic variation among populations, pairwise $F_{ST}$ values and analysis of molecular variance (AMOVA) were computed using Arlequin v3.5 [28]. For the hierarchical analysis, populations were grouped and defined based on geographical factors: (1) two geographical groups, Pacific coast and other, were primarily divided by the Kuroshio Current; (2) three geographical groups, Ryukyu Islands, Taiwan, mainland China (Taiwan Strait), and Japan, were primarily divided by the ocean region; and (3) five geographical groups, Ryukyu Islands, Taiwan Island, mainland China, Japan, and Korea, were primarily divided by the sampling region.

Phylogenetic relationships were reconstructed using neighbor-joining (NJ), maximum likelihood (ML), and Bayesian inference (BI) methods, implemented in MEGA-X [29], PhyML 3.0 [30], and MrBayes v3.1.2 [31], respectively. The best-fit nucleotide substitution model was determined using Smart Model Selection with the Akaike Information Criterion (AIC) [32] in PhyML 3.0, and the GTR + G + I model [33] was selected as the best nucleotide replacement model. We conducted BI analyses with two concurrent runs of approximately 1,500,000 iterations, with samples recorded every 5000 iterations. The first 25% of trees were discarded as burn-in, and the remaining trees were used to generate a 50% majority rule consensus. For the ML analysis, we used the GTR + G + I substitution model for the nucleotide dataset, and bootstrap resampling was performed using the rapid option with 1000 iterations. Branch support was evaluated with 100 pseudoreplicates. For NJ analysis, we applied the Kimura 2-parameter model, and bootstrapping was performed with 1000 replications. Haplotype networks were constructed using the MINSPNET algorithm implemented in Popart [34].

### 2.3. Historical Demography, Divergence Time Estimation and Biogeographic Analysis

To test the assumption of neutrality, we analyzed historical demographic scenarios using Tajima's D test [35] and Fu's Fs test [36], and computed mismatch distributions in DnaSP v5.0 [26]. Bayesian skyline plot (BSP) analyses were performed in BEAST v1.8.2 for each lineage in *M. labio* [37] to determine effective population size changes over time. To ensure convergence of all parameters (ESSs > 200), we ran >200,000,000 MCMC iterations and discarded the first 10% of samples for each chain as burn-in. Plots for each analysis were generated using Tracer v1.6 [38]. Additionally, we used BEAST v1.8.2 software [37] to calculate the time to the most recent common ancestor (TMRCA). In this study, we adopted a mutation rate of 2.4% per million years calculated from two *Tegula* species separated by the Isthmus of Panama and calibrated for the mtDNA COI genes for population expansion [17,39]. We utilized MIGRATE v3.2.6 to ascertain the historical migration rate direction (M = m/$\mu$, where m = migration rate and $\mu$ = the per-locus mutation rate) between Taiwan, the Ryukyu Islands, and Japan (including GJ) [40]. Maximum likelihood inference was performed in MIGRATE, comparing a full migration model (where $\theta$ and M were estimated jointly from the data) to a restricted model (where $\theta$ was averaged and M was assumed to be symmetrical among populations). Each run included five replicates of ten short chains (with 10,000 genealogies sampled) and three long chains (with 100,000 genealogies sampled), discarding the first 10,000 genealogies. The model was run five times to ensure parameter estimates' convergence, assuming a migration matrix model with unequal population sizes and different migration rates [40].

We employed approximate Bayesian Computation (ABC) methods with DIYABC v.2.0.4 [41] to investigate the potential historical demographic scenarios of *M. labio* and *M. confusa*. We defined five demographic scenarios, as suggested by Cabrera and Palsboll [42] (Figure 2), to explore the demographic histories of *M. labio* and *M. confusa*. The five scenarios were as follows: (1) a constant size model (CON); (2) a bottleneck event

model (DEC); (3) a recent expansion model (INC); (4) an expansion followed by a decrease model (INCDEC); and (5) a decrease followed by an expansion model (DECINC). We generated three million datasets for each scenario and calculated summary statistics using DIYABC [40]. The posterior probability of each scenario was obtained using both the direct approach and the logistic regression approach implemented in DIYABC [40].

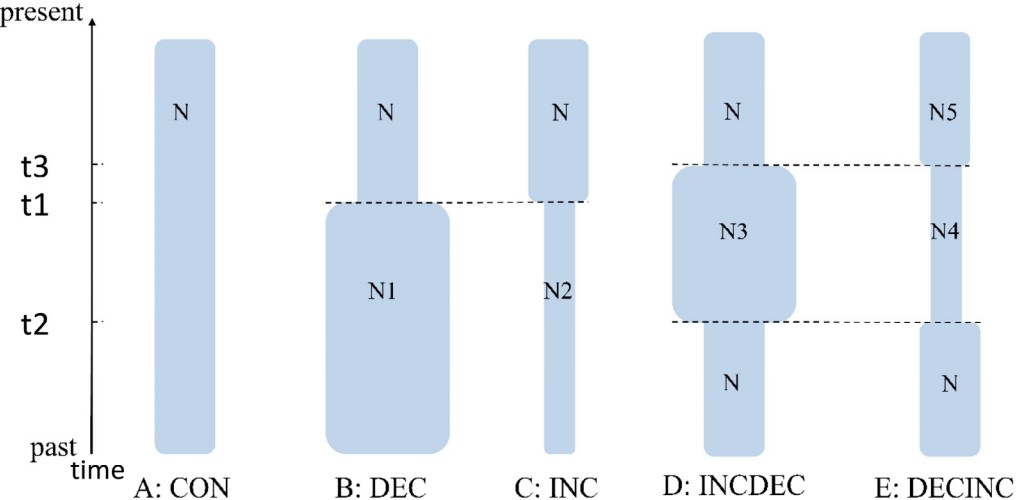

**Figure 2.** Schematic representation of five demographic scenarios for *Monodonta labio* and *M. confusa* tested by approximate Bayesian computation (ABC) Time (**t1–t3**) and effective population size (N–N5) are not scaled.

## 3. Results

### 3.1. Genetic Diversity

A total of 1340 base pairs (bp) of mtDNA COI + 16S gene sequences from 85 specimens were analyzed and found to contain 25.0% adenine, 32.1% thymine, 20.7% cytosine, and 22.1% guanine (42.8% GC content) in *M. labio*. Out of the 41 specimens analyzed in *M. confusa*, the DNA composition revealed 25.2% adenine, 32.4% thymine, 20.3% cytosine, and 22.1% guanine, resulting in a GC content of 42.4%. The lengths of the COI and 16S markers were 769 and 571 bp, respectively. A total of 61 haplotypes were identified, and the average haplotype diversity was high (0.987), ranging from 0.901 (SY) to 1.000 (AD, WW, MZ, and GJ) in *M. labio*. In *M. confusa*, we identified a total of 33 haplotypes, and the average haplotype diversity was high at 0.983, ranging from 0.867 (CF) to 1.000 (AD, TD, and YG). The average nucleotide diversity ($\theta\pi$) was low (0.0131), ranging from 0.0012 (YG) to 0.0042 (GY) in *M. labio* (Table 1). In *M. confusa*, the average nucleotide diversity ($\theta\pi$) was found to be low at 0.0082, with values ranging from 0.0031 (CF) to 0.0093 (AD), as shown in Table 1. There were only six haplotypes that occurred in more than one population, with one haplotype (H29) observed in five populations and the others only shared by specimens from two populations of *M. labio* (Table S1). In *M. confusa*, only four haplotypes were found to occur in more than one population, being shared by specimens from two populations (Table S1). Only two populations (OK and SY) in the Ryukyu Islands do not share any haplotypes with other populations of *M. labio*.

### 3.2. Phylogenetic Reconstruction and Genetic Structure

The significant correlation between phylogeny and geography was evident through the considerably larger $N_{ST}$ values compared to $G_{ST}$ (0.914 and 0.035 in *M. labio*, 0.266 and 0.026 in *M. confusa*, respectively), consistent with findings from Pons and Petit [43]. Using COI + 16S for phylogenetic analysis, we identified three main lineages (IA, IB, and IC) in *M. labio*. Lineage IA was exclusively found in populations OK and SY in the Ryukyu Islands. Lineage IB consisted of individuals from populations in Taiwan and mainland China, while lineage IC was distributed among populations from Taiwan Island,

mainland China, Japan, and one population (GJ) from Korea. Based on the phylogenetic tree, three main lineages (IIA, IIB, and IIC) were identified in *M. confusa*. Lineages IIA and IIB were present in populations from Taiwan Island and mainland China, while lineage IIC exclusively occurred in populations from Japan. The NJ, ML, and BI methods yielded the same tree topologies in the phylogenetic analysis (Figure 3), which supported the haplotype minimum-spanning network topology and showed no clustering corresponding to the sampling site (Figure 4). The topological relationships obtained from the phylogenetic analysis with other sequences of *M. labio* and *M. confusa* from GenBank also support the formation of two major lineages (Figure S1).

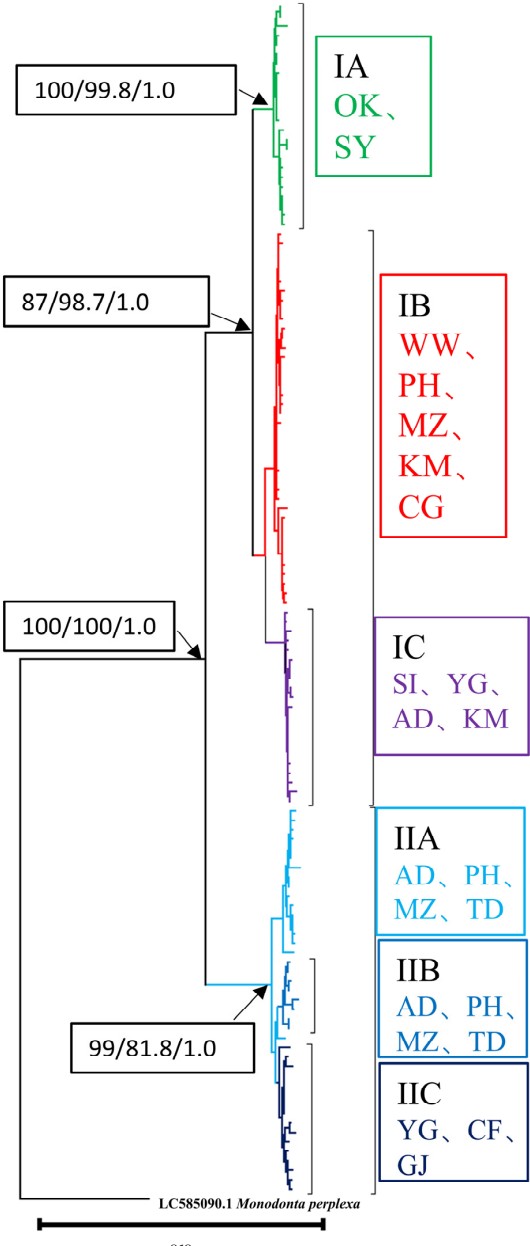

**Figure 3.** Reconstructed phylogeny of *Monodonta labio* and *M. confusa* using neighbor-joining (NJ) analyses based on the mitochondrial COI + 16S gene among 13 populations using 94 haplotypes The numbers above each branch are the posterior probabilities for bootstrap values for the NJ, ML, and Bayesian analyses.

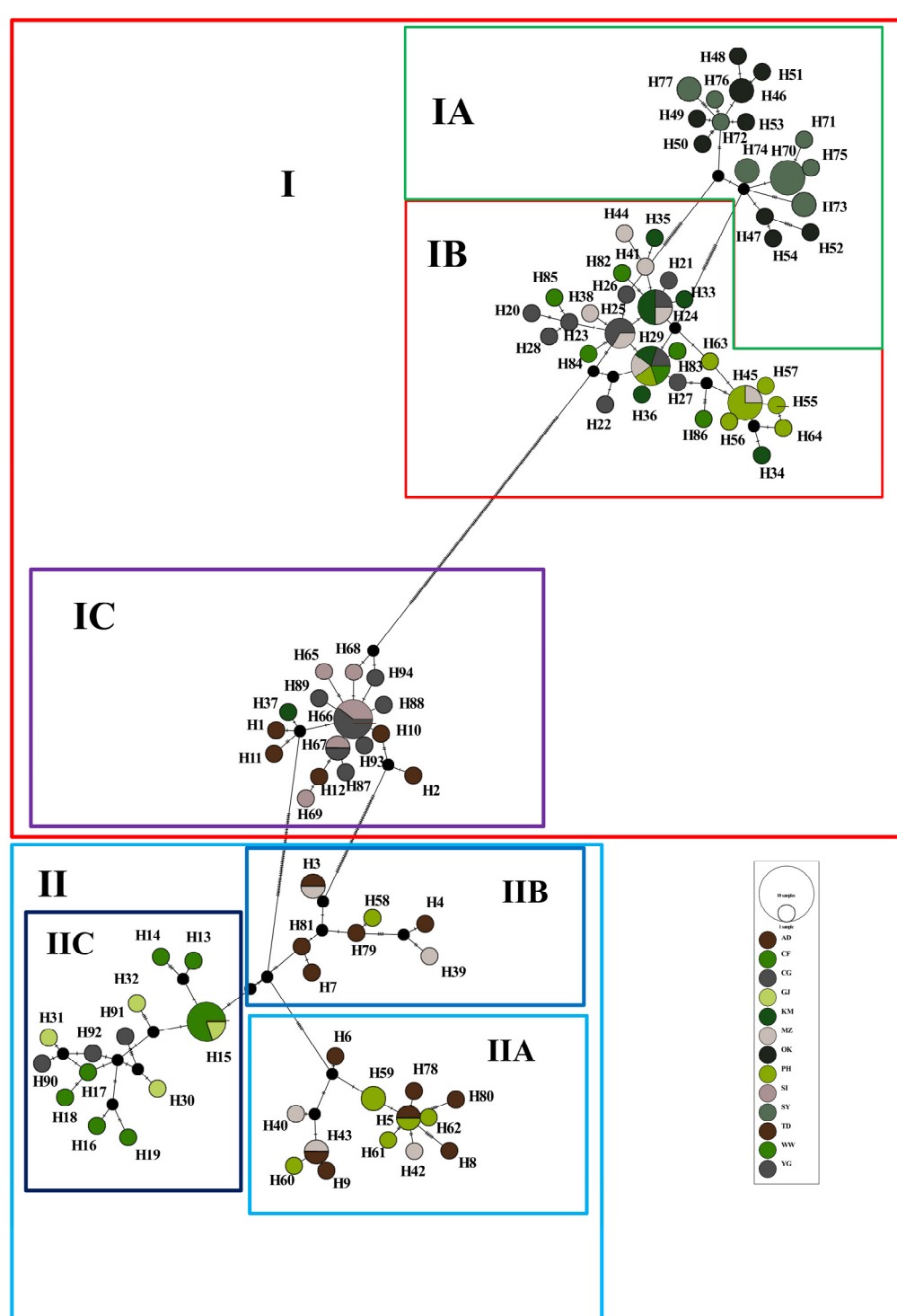

**Figure 4.** The median-joining network of 94 haplotypes of *Monodonta labio* and *M. confusa*. Each circle represents a haplotype, and the circle size is proportional to the haplotype frequency. The bars across the branches are the number of mutational steps. Black circles represent missing haplotypes. The sub-figures (I, II, A, B, C) referred to in this figure correspond to the clusters in Figure 3.

The range of pairwise $F_{ST}$ values between populations ranged from −0.047 (between MZ and WW) to 0.985 (between SI and PH), with a mean value of 0.911 in *M. labio*. In *M. confusa*, the pairwise $F_{ST}$ values between populations varied from −0.138 (between MZ and TD) to 0.605 (between CF and PH), with an average value of 0.265. Moreover, the pairwise $F_{ST}$ values between populations from different regions were relatively high and

significant, indicating that high differentiation existed among these populations (Table 2). The AMOVA results indicate significant genetic structure at scenarios II and III in *M. labio* and *M. confusa* (Table 3). The results from the five geographic regions (Scenario III: Taiwan, mainland China, Ryukyu Islands, Japan, and Korea) demonstrated that significant spatial genetic structuring among groups was 59.93% ($F_{CT}$ = 0.599, $p$ < 0.001) but was 31.74% ($F_{SC}$ = 0.792, $p$ < 0.001) among populations within groups and 8.31% ($F_{ST}$ = 0.916, $p$ < 0.001) within populations in *M. labio* (Table 3). Furthermore, the findings from the five geographic regions (Scenario III) revealed significant spatial genetic structuring among groups at 33.51% ($F_{CT}$ = 0.335, $p$ < 0.001). The genetic structuring among populations within groups was minimal at −0.27% ($F_{SC}$ = −0.004, $p$ = 0.546), with the majority of the variation, 66.75%, being attributed to within-population differences ($F_{ST}$ = 0.332, $p$ < 0.001) in *M. confusa* (Table 3). However, AMOVA from the two geographic regions (Pacific Coast and others) identified only −5.50% and −10.46% of the variants to be present among groups, 72.41% and 35.55% of the variation among populations within groups, and 33.10% and 74.90% of the variation within populations in *M. labio* and *M. confusa*, respectively (Table 3).

**Table 2.** (**A**). Matrix of pairwise $F_{ST}$ among thirteen populations based on mitochondrial COI + 16S gene (below diagonal) and *p* values (above diagonal) in *Monodonta labio*. (**B**). Matrix of pairwise $F_{ST}$ among thirteen populations based on mitochondrial COI + 16S gene (below diagonal) and *p* values (above diagonal) in *Monodonta confusa*.

| (A) | | | | | | | | | | |
|---|---|---|---|---|---|---|---|---|---|---|
| | **AD** | **WW** | **CG** | **PH** | **KM** | **MZ** | **YG** | **SI** | **OK** | **SY** |
| AD | | 0.000 | 0.000 | 0.000 | 0.000 | 0.000 | 0.000 | 0.270 | 0.000 | 0.000 |
| WW | 0.968 | | 0.414 | 0.000 | 0.891 | 0.774 | 0.000 | 0.000 | 0.000 | 0.000 |
| CG | 0.980 | 0.000 | | 0.000 | 0.252 | 0.513 | 0.000 | 0.000 | 0.000 | 0.000 |
| PH | 0.981 | 0.455 | 0.594 | | 0.000 | 0.000 | 0.000 | 0.000 | 0.000 | 0.000 |
| KM | 0.799 | −0.041 | 0.034 | 0.145 | | 0.981 | 0.000 | 0.000 | 0.000 | 0.000 |
| MZ | 0.975 | −0.047 | −0.008 | 0.519 | −0.030 | | 0.000 | 0.000 | 0.000 | 0.000 |
| YG | 0.152 | 0.979 | 0.986 | 0.987 | 0.849 | 0.984 | | 0.558 | 0.000 | 0.000 |
| SI | 0.029 | 0.974 | 0.983 | 0.985 | 0.816 | 0.980 | −0.015 | | 0.000 | 0.000 |
| OK | 0.973 | 0.795 | 0.849 | 0.850 | 0.483 | 0.819 | 0.981 | 0.977 | | 0.000 |
| SY | 0.975 | 0.797 | 0.843 | 0.844 | 0.522 | 0.817 | 0.981 | 0.978 | 0.154 | |

| (B) | | | | | | | |
|---|---|---|---|---|---|---|---|
| | **AD** | **TD** | **PH** | **MZ** | **CF** | **YG** | **GJ** |
| AD | | 0.972 | 0.396 | 0.936 | 0.000 | 0.036 | 0.027 |
| TD | −0.133 | | 0.594 | 0.990 | 0.000 | 0.009 | 0.072 |
| PH | 0.013 | −0.042 | | 0.342 | 0.000 | 0.000 | 0.009 |
| MZ | −0.137 | −0.138 | −0.002 | | 0.000 | 0.036 | 0.036 |
| CF | 0.395 | 0.442 | 0.605 | 0.459 | | 0.036 | 0.279 |
| YG | 0.332 | 0.351 | 0.053 | 0.356 | 0.193 | | 0.576 |
| GJ | 0.260 | 0.284 | 0.528 | 0.306 | 0.018 | −0.049 | |

Estimates of gene flow calculated with the MIGRATE program indicated the possibility of genetic exchange among the southern region (Taiwan and mainland China group), the central region (Ryukyu Islands group), and the northern region (Japan and Korea group). Asymmetrical gene flow among the three regions was found, indicating that individuals from the southern region may have colonized the northern region. The estimate of historical gene flow from the northern region to the southern region and the southern region to the central region was larger in *M. labio* (M $_{S \to N}$ = 215.1, 97.5% CI: 78~328) and M $_{C \to N}$ = 101.8, 97.5% CI: 25.3~184.0, where the suffixes S, C, and N indicate the southern region, the central region, and the northern region, respectively. In *M. confusa*, the estimation of historical gene flow indicated similarity in gene flow from the northern region to the southern region and from the southern region to the northern region.

**Table 3.** Analysis of molecular variance (AMOVA) for *Monodonta labio* and *M. confusa* populations based on mitochondrial COI + 16S genes.

| Source of Variation | Percentage Variation | | *p* |
|---|---|---|---|
| *Monodonta labio* | | | |
| 1. Scenario I: two groups (WW, CG, PH, KM, MZ, YG) (AD, OK, SY, SI) | | | |
| Among groups | −5.50 | $F_{CT} = -0.055$ | 0.461 |
| Among populations within groups | 72.41 | $F_{SC} = 0.686$ | 0.000 |
| Among individuals within populations | 33.10 | $F_{ST} = 0.669$ | 0.000 |
| 2. Scenario II: three groups (AD, WW, CG, PH, KM, MZ) (OK, SY) (YG, SI) | | | |
| Among groups | 68.40 | $F_{CT} = 0.683$ | 0.016 |
| Among populations within groups | 24.36 | $F_{SC} = 0.770$ | 0.000 |
| Among individuals within populations | 7.24 | $F_{ST} = 0.927$ | 0.000 |
| 3. Scenario III: five groups (AD, WW, CG, PH) (KM, MZ) (OK, SY) (YG, SI) | | | |
| Among groups | 59.93 | $F_{CT} = 0.599$ | 0.000 |
| Among populations within groups | 31.74 | $F_{SC} = 0.792$ | 0.000 |
| Among individuals within populations | 8.31 | $F_{ST} = 0.916$ | 0.000 |
| *Monodonta confusa* | | | |
| 1. Scenario I: two groups (PH, MZ, YG, GJ) (AD, TD, CF) | | | |
| Among groups | −10.46 | $F_{CT} = -0.104$ | 1.000 |
| Among populations within groups | 35.55 | $F_{SC} = 0.321$ | 0.000 |
| Among individuals within populations | 74.90 | $F_{ST} = 0.250$ | 0.000 |
| 2. Scenario II: three groups (AD, TD, PH, MZ) (CF, YG, GJ) | | | |
| Among groups | 43.25 | $F_{CT} = 0.432$ | 0.032 |
| Among populations within groups | −2.04 | $F_{SC} = -0.035$ | 0.686 |
| Among individuals within populations | 58.78 | $F_{ST} = 0.412$ | 0.000 |
| 3. Scenario III: five groups (AD, TD, PH) (MZ) (CF, YG) (GJ) | | | |
| Among groups | 33.51 | $F_{CT} = 0.335$ | 0.000 |
| Among populations within groups | −0.27 | $F_{SC} = -0.004$ | 0.546 |
| Among individuals within populations | 66.75 | $F_{ST} = 0.332$ | 0.000 |

### 3.3. Molecular Dating and Historical Demographic Expansion

The estimated Tajima's D and Fu's *F*s indexes were not consistent in the overall population of *M. labio*, with positive Tajima's D (1.51946) and significantly negative Fu's *F*s (−1.931). In *M. confusa*, the estimated Tajima's D and Fu's *F*s indexes showed inconsistency. Tajima's D was nonsignificantly negative (−1.16158), while Fu's *F*s was significantly negative (−15.823). Multiple-modal mismatch distributions were observed in the total population, indicating no recent demographic expansion for the populations of *M. labio* and *M. confusa*. The Bayesian skyline plot analysis, which assumed a substitution rate of 2.4% per million years, revealed that *M. labio* populations underwent a population decline followed by rapid population growth around 0.1 million years ago, as shown in Figure 5a. Figure 5b indicates that populations of *M. confusa* experienced a population growth event around 0.075 million years ago. These results showed that the history of population size changes in *M. labio* and *M. confusa* is complex. We used the five scenarios of ABC analysis to determine the possible demographic history of *M. labio*. In the ABC modeling, Scenario 5 (DECINC model) was highly favored (posterior probability = 1.0000 [1.0000, 1.0000]) compared with the other scenarios in *M. labio*. In *M. confusa*, Scenario 1 (CON) was highly favored (posterior probability = 0.9698 [0.9613, 0.9783]) compared with the other scenarios. This scenario suggested that *M. labio* experienced a reduction in the effective population size in the past, followed by a single instantaneous increase in population size. The estimated time to coalescence in *M. labio* was estimated at 2.885 ± 0.441 myr based on results obtained with BEAST. The estimated time to the most recent common ancestor (TMRCA) of *M. confusa* is approximately 0.286 ± 0.103 myr.

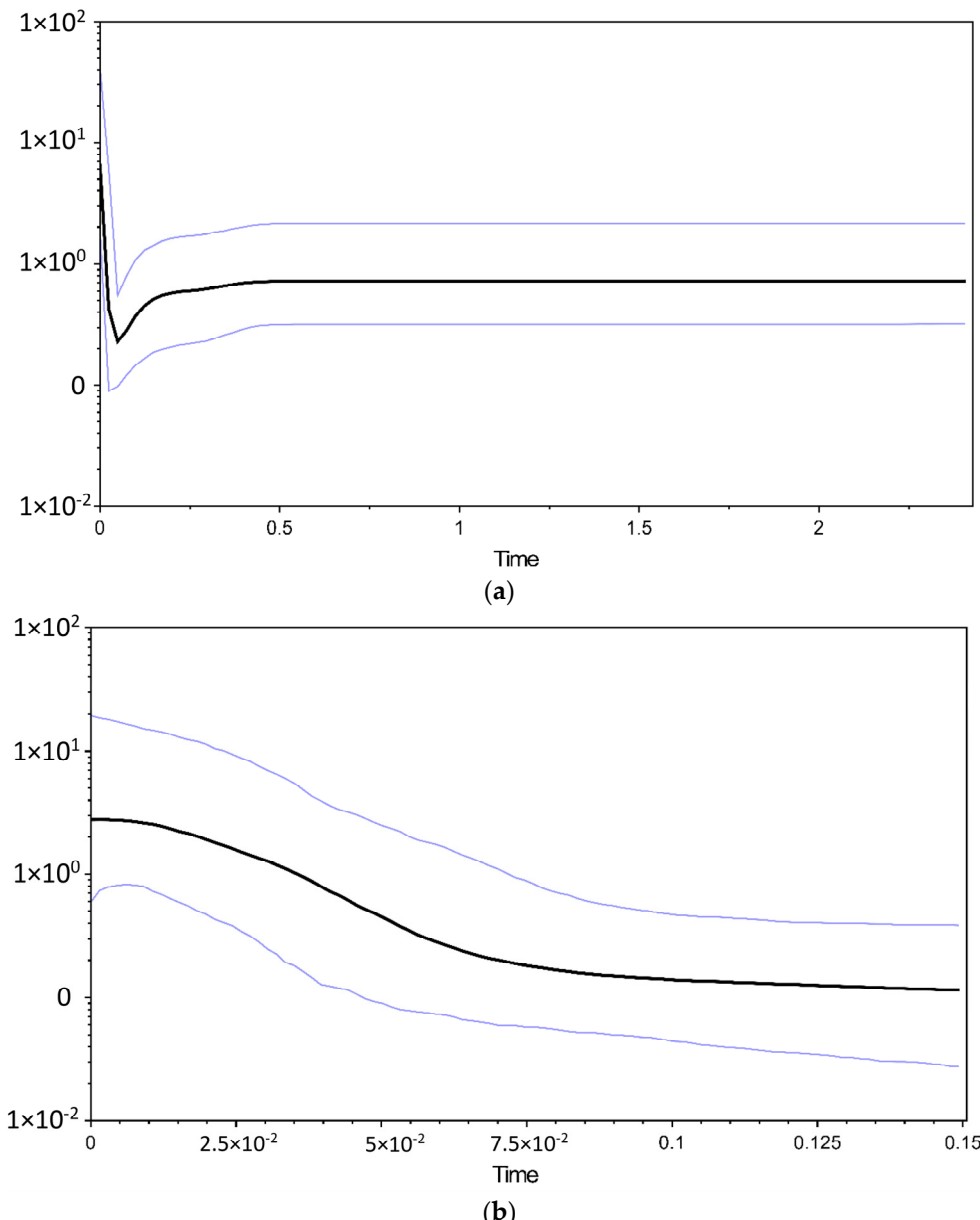

**Figure 5.** Bayesian skyline plot of the effective population sizes over time for *Monodonta labio* (**a**) and *M. confusa* (**b**) The units of the X axis are millions of years, and the units of the Y axis are Ne (the log-transformed product of effective population size and generation length in years).

## 4. Discussion

### 4.1. Genetic Diversity

Maintaining genetic diversity is crucial for populations to persist and enable them to adapt and evolve in response to environmental changes. Several factors can affect the genetic diversity of marine organisms, including overfishing, selection, pollution, the presence of dispersal barriers, dispersal capacity, genetic drift within the population, and inbreeding. In this study, analyzing the genetic diversity of *M. labio* and *M. confusa* populations revealed higher haplotype diversity (0.987, 0.983) and nucleotide diversity (0.0131, 0.0082) than previous studies conducted in mainland China and Japan (e.g., [20–22]). In our results, the remarkably lower nucleotide diversity may be due to the lower effective population size of *M. labio* and *M. confusa* in the Ryukyu Islands and Japan, as compared to that of Taiwan and mainland China (Table 1). *Monodonta labio* and *M. confusa* are widely distributed along the coast of East Asia. In our studies, populations in Taiwan and

mainland China at the center of the distribution are likely to have high genetic diversity due to gene flow from neighboring populations in the Ryukyu Islands and Japan that are better adapted to different conditions. The combination of low nucleotide diversity and high haplotype diversity has often been linked to founder effects or population bottlenecks [43]. In previous studies, *M. labio* was identified as a habitat specialist, whereas *M. confusa* showed no habitat specificity. Additionally, *M. labio* exhibited lower genetic diversity and higher genetic differentiation among populations when compared to *M. confusa*. In our studies, we observed a comparable pattern of nucleotide diversity in *M. labio*, which showed higher nucleotide diversity than *M. confusa*. This suggests that our study had a larger sampling range compared to previous studies [22]. However, the abundance and uniform distribution of private haplotypes within *M. labio* and *M. confusa* populations indicate a large effective population size, likely due to their wide distribution throughout East Asia.

*4.2. Population Structure and Demographic History*

Pairwise $F_{ST}$ values suggested genetic differentiation was substantially greater among populations between regions than among populations within regions. Furthermore, our AMOVA and IBD analyses among populations of *M. labio* and *M. confusa* suggested lower proportions of genetic admixture. The AMOVA results revealed that the genetic differentiation of the total genetic variation among groups in *M. labio* and *M. confusa* from the three geographic regions (Scenario II) accounted for 68.40% ($F_{CT} = 0.683$, $p = 0.016$) and 43.25% ($F_{CT} = 0.432$, $p = 0.032$), respectively. However, the variation among populations within groups was 24.36% ($F_{SC} = 0.770$, $p < 0.000$) for *M. labio* and $-2.04$% ($F_{SC} = 0.035$, $p = 0.686$) for *M. confusa*, while the variation within populations was 7.24% ($F_{ST} = 0.927$, $p < 0.001$) for *M. labio* and 66.75% ($F_{ST} = 0.332$, $p < 0.000$) for *M. confusa* (Table 2). These results were consistent with our IBD results. The life cycle of *M. labio* includes a relatively short larval stage, which might result in genetic differentiation [22]. The findings of our study align with previous research, indicating that *M. labio* displays higher genetic differentiation among populations than *M. confusa* [22]. The higher genetic differentiation observed in *M. labio* is consistent with previous studies that reported significant differences between populations in Japan and mainland China [20–22]. Yamazaki et al. [22] indicated that the genetic structure of *M. labio* and *M. confusa* seems to be strongly influenced by geographical factors, such as the dispersal barrier between the Japanese Islands and the Ryukyu Islands. In our study, lineage IA was a monophyletic group only distributed in the Ryukyu Islands. We suggested *that M. labio* has relatively higher genetic differentiation and lower nucleotide diversity in the Ryukyu Islands, which have experienced significant founder effects and bottlenecks. Within *M. confusa*, lineage IIC constituted a monophyletic group solely distributed in the Japan Islands, implying relatively higher genetic differentiation between Taiwan Island, mainland China, and Japan.

In general, genetic differentiation among populations is influenced by their dispersal abilities in marine invertebrate populations [44]. The results from MIGRATE suggested asymmetrical gene flow, with higher migration rates from the southern and central regions to the northern region. Such a scenario would occur in the context of the biological characteristics and population demographics of *M. labio*. In the spring, *M. labio* lays its eggs in shallow coastal waters and exhibits a planktonic larval stage that facilitates dispersal, following the coastal current that flows northeastward through the NWP marginal seas in spring and summer [45]. Kuroshio Current systems probably played a more predominant role in shaping the genetic homogeneity among populations of *M. labio* in NWP marginal seas [22]. The influence of combined southward-running currents (the China Coastal Current (CCC)) on the population genetic connectivity of *M. labio* is presumably weaker than the solely northward-flowing Kuroshio Current. The results confirm that the dynamic system associated with the variation in dispersal ability may be affected by the dominant Kuroshio current systems of the Pacific Ocean. Such a scenario might apply to other marine species in NWP marginal seas, such as *Sicyopterus japonicas* [15] and

*Sargassum fusiforme* [46]. In ecological terms, *M. confusa* demonstrates a broad tolerance for different wave exposures and exhibits no specific habitat preference. Regarding population differentiation, *M. confusa* shows lower levels compared to *M. labio*. Based on the results from the MIGRATE analysis, the historical gene flow estimation indicates similarity in gene flow between populations. Therefore, we conclude that the complex currents in the Western Pacific consistently influence the dispersal of *M. confusa*.

Neutrality indices by Tajima's D tests and multi-modal mismatch distribution analyses suggested that *M. labio* and *M. confusa* populations indicate no recent demographic expansion, but the significantly negative value of Fu's *F*s tests indicated recent demographic expansion. In general, Fu's *F*s tests are more sensitive at detecting demographic events than Tajima's D tests. However, high haplotype diversity and low nucleotide diversity were found in the population, and the results of Bayesian skyline plot analysis were consistent with a recent population expansion [35,36,47]. The Bayesian skyline plot analyses unveiled a late Pleistocene demographic expansion for *M. labio*, which commenced approximately 0.1 million years ago. Additionally, *M. confusa* populations exhibited a population growth event around 0.075 million years ago. Similar results were observed in previous studies of *M. labio* and *M. confusa* in mainland China, as well as in other marine organisms such as *S. macrorhynchos* [48] and *Nemipterus bathybius* [49]. The demographic history of *M. labio* and *M. confusa* is more intricate, as evidenced by our approximate Bayesian Computation (ABC) analysis. The results suggested that all populations of *M. labio* experienced a decline in population size after a recent expansion, while *M. confusa* populations remained constant in size.

### 4.3. Phylogeography of Monodonta labio

Yamazaki et al. [21] noted that the genus *Monodonta* undergoes a planktonic stage in their lifecycle, and the planktonic period is reported as three days, revealing that species were likely affected by geographic isolation with regard to their intraspecific divergence, population structure, and geographical distribution. Tempo-spatial genetic architecture is observed between two mitochondrial lineages based on the haplotype network and phylogenetic analysis of *M. labio* and *M. confusa*. We incorporated sequences published by other studies into our analysis and found that they can be divided into two groups, lineages I (*M. labio*) and II (*M. confusa*). Among the individuals in the lineage II group, Zhao et al. [20] identified them as *M. labio* (refer to "clade D" in Zhao et al. [20]), while Yamazaki et al. [20,21] identified them as *M. confusa* (refer to "clade D" in Yamazaki et al. [20,21]). Due to the wide distribution range of this group, in our study, to avoid confusion in the classification system and to focus on the population's historical dynamics, we agree with the viewpoint of Yamazaki et al. [21] and consider the individuals in lineage II as *M. confusa* (Figure S1, Table S2). Historically, many phylogeographic scenarios for marine organisms have responded to sea-level fluctuations during Pleistocene glaciations. The Taiwan Strait acted as a land bridge due to the fact that the sea level fell to 130 m below the present sea level (generally shallower than 60 m) within glacial periods [50]. The estimated time to the most recent common ancestor (TMRCA) of lineages I (*M. labio*) and II (*M. confusa*) is estimated at 2.885 myr and 0.441 myr, respectively; however, no direct evidence demonstrates that the Taiwan Strait Land Bridge, which acted as a physical barrier for the *M. labio* and *M. confusa* populations, existed during the period [20]. Many previous studies of some marine organisms suggested that two distinct genetic lineages were consistent with our findings in the Northwest Pacific Ocean, such as *Macrobrachium nipponense* [51], *Terapon jarbua* [52], Chinese black sleeper (*Bostrychus sinensis*) [53], and spotted scat (*S. argus*) [6]. The lineage I (*M. labio*) was subdivided into three sub-lineages (IA, IB, and IC): the Ryukyu Islands and the populations in Taiwan, mainland China, and Japan. The same pattern of genetic differentiation in the Ryukyu Islands has previously been observed in the same species [20,21]. In other studies, individuals of sub-lineage IA were also found to be distributed in Vietnam and the Gulf of Tonkin region (Figure S1) [20]. Sub-lineages IB are mainly distributed in Taiwan, Penghu, and mainland China, while

sub-lineages IC are mainly distributed in Japan and South Korea [20–22]. Lineages II (*M. confusa*) have the widest distribution in the Northwest Pacific region, with distribution across various regions including Japan, South Korea, mainland China, and Taiwan, except for the Ryukyu Islands (Figure S1, Table S2) [20–22]. In previous studies of *M. confusa*, the research locations were primarily focused on Japan and the Ryukyu Islands, without including Taiwan and mainland China. The genetic differentiation between Japan and the Ryukyu Islands was not significant. In our study, among the three sub-lineages, two (IIA and IIB) are mainly distributed in Taiwan and mainland China, while the IIC sub-lineage is primarily distributed in Japan. This indicates a certain level of differentiation among the sub-lineages, which we believe is a result of distance and ocean currents. We suggested that the specialist-generalist variation hypothesis (SGVH) is the most important factor for genetic differentiation and that the Kuroshio Current is an effective influencer of the genetic differentiation of *M. labio*, supporting previous studies [21,54].

**Supplementary Materials:** The following supporting information can be downloaded at: https://www.mdpi.com/article/10.3390/d15091021/s1. Table S1: The distribution information of the shared haplotypes (Hm1–Hm 94) (COI + 16S) *p* indicates the number of private haplotypes in each population. Table S2: The haplotype information downloaded from NCBI. Figure S1: The phylogeny of *Monodonta labio* (Lineage I) and *M. confusa* (Lineage II) was reconstructed using neighbor-joining (NJ) analyses based on the mitochondrial COI + 16S gene. This analysis involved a total of 126 haplotypes, including 68 sequences obtained from NCBI and 94 haplotypes derived from our study. The numbers above each branch are the posterior probabilities for bootstrap values for the NJ, ML, and Bayesian analyses. The haplotypes marked in black on the phylogenetic tree are haplotypes downloaded from NCBI.

**Author Contributions:** Conceptualization, Y.-W.C. and B.-S.S.; methodology, H.B.; software, H.B. and J.-X.W.; validation, H.B., B.-S.S. and J.-X.W.; investigation, Y.-W.C. and H.B.; resources, Y.-W.C.; data curation, Y.-W.C., H.B. and H.-D.L.; writing—original draft preparation, Y.-W.C. and H.-D.L.; writing—review and editing, Y.-W.C. and H.-D.L. project administration, Y.-W.C.; funding acquisition, Y.-W.C. All authors have read and agreed to the published version of the manuscript.

**Funding:** This research was funded by National Science and Technology Council of Taiwan, MOST 110-2621-B-415-004 and MOST 105-2621-B-006-005.

**Institutional Review Board Statement:** All animal experiments were conducted in the laboratory of Yuh-Wen Chiu at Nation-al Chiayi University, following the guidelines and approval of the Animal Research and Ethics Committee of the same university.

**Data Availability Statement:** All sequences have been deposited in GenBank under the following accession numbers: OQ880826-OQ 880919.

**Acknowledgments:** We thank Da-Ji Huang for his assistance in the field and help with the sample collection. Thanks also to Shih-Hsiung Liang for their invaluable review of and improvements to our manuscript. We also thank the anonymous reviewers and associate editor for their helpful comments.

**Conflicts of Interest:** The authors declare no conflict of interest.

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
