# Peer review of "Population Structure and Phylogeography of Marine Gastropods Monodonta labio and M. confusa (Trochidae) along the Northwestern Pacific Coast"

_diversity, doi:10.3390/d15091021_

Round 1

Reviewer 1 Report

The manuscript uses DNA sequences from two mitochondrial genes to study the population structure and phylogeography of the  intertidal gastropod, Monodonta labio. Previous articles have also examined this subject but have had a more restricted geographic focus. The present manuscript is based on the collection of data from areas that were previously understudied including Taiwan. It is however, severely limited by the omission of integration of the present work with previous studies. In particular the two clades designated here should be related to the clades identified in Zhao et al. (citation 20 in the article) and Yamazaki et al. (citation 22).  Ideally this should be done by combined analysis of the sequences available in GenBank. However, even in the absence of such an analysis, it should be possible to make a reasonable hypothesis about the correspondence of clade designations. This would allow discussion of the complete range of Monodonta labio from East Asia that have been sequenced. This range extends further south-west and north along the coasts of mainland China than the samples collected here.

The topology used as an illustration of the phylogenetic analysis is based on the neighbour joining model. It raises questions about the relationships within the analysed sequences. It should be stated whether these are consistently observed in other analyses. Group IB is not a lineage per se as the sister group of lineage IA is only a part of the group. Lineage II lacks statistical support values – I assume this is simply an omission. What is the status of the large number of sequences in this lineage that are excluded from the strongly supported derived clade? Is it a previously recognised clade that has not been distinguished in this study?

The genetic distances between lineages I and II are apparently very high for conspecific taxa (>0.10 estimating from Figure 3). It would be desirable to check that the lineages are not cryptic species using a nuclear gene sequence (for example ITS-1 or 2, or a nuclear gene intron) but if this is not possible then parallel series of analyses of population genetic and demographic parameters should be performed for Lineages I and II and any notable differences between the series reported in the text. I note that this has already been done for the AMOVA (two group model).

The historical demography analyses do not do much to clarify the population history of Monodonta labio in the region with the Bayesian Skyline Plot suggesting a recent size expansion after a previous decline and the Approximate Bayesian Computation suggesting a recent decline after a previous expansion. Size changes might occur if climate changes are combined with barriers to gene during glacial maxima but there have been numerous cycles during the Plio-Pleistocene and it is unclear that the approaches used here can effectively model this. It should be acknowledged in the text that the models used are rather simplistic. Given the disagreement between analyses, causal attributions (e.g. lines 371–373) should be omitted as being merely speculative.

In section 4.3, a date of 0.462 Ma is given for divergence between lineages I and II it is unclear how this is derived (and appears too low if the scale in Figure 3 is correctly labelled).

(line 349) Is there any information as to how long the Kuroshio current has been flowing, and at what strength?

GenBank accession numbers need to be added.

The writing is generally clear and correct, with occasionally odd word choices that could be revised later.  It is worth noting that the “Ryukyu Islands” are incorrectly referred to as “Ryukyu Island” in the text.  I suggest “investigate” rather than “act” on line 43. The spelling should be “multi-modal” on line 358. “imply that” should be “apply to” or similar (line 356).

Author Response

The manuscript uses DNA sequences from two mitochondrial genes to study the population structure and phylogeography of the  intertidal gastropod, Monodonta labio. Previous articles have also examined this subject but have had a more restricted geographic focus. The present manuscript is based on the collection of data from areas that were previously understudied including Taiwan. It is however, severely limited by the omission of integration of the present work with previous studies. In particular the two clades designated here should be related to the clades identified in Zhao et al. (citation 20 in the article) and Yamazaki et al. (citation 22).  Ideally this should be done by combined analysis of the sequences available in GenBank. However, even in the absence of such an analysis, it should be possible to make a reasonable hypothesis about the correspondence of clade designations. This would allow discussion of the complete range of Monodonta labio from East Asia that have been sequenced. This range extends further south-west and north along the coasts of mainland China than the samples collected here.

Response: As requested, we downloaded the relevant sequences from GenBank, including those from Zhao et al.'s study (reference 20 in the paper) and Yamazaki et al.'s study (reference 22). We reconstructed the phylogenetic tree and discussed the relationships between different populations.

The topology used as an illustration of the phylogenetic analysis is based on the neighbour joining model. It raises questions about the relationships within the analysed sequences. It should be stated whether these are consistently observed in other analyses.

Response: We constructed phylogenetic trees using three methods: NJ, ML, and Bayesian analyses. The trees generated by these three methods were largely consistent with each other. Therefore, we annotated bootstrap values at important nodes.

Group IB is not a lineage per se as the sister group of lineage IA is only a part of the group.

Response: As requested, we incorporated the suggestions from other reviewers and included an outgroup analysis in our phylogenetic tree. Upon adding the outgroup, we observed that Group IB formed a monophyletic group.

Lineage II lacks statistical support values – I assume this is simply an omission.

Response: As requested, we revised and added it.

What is the status of the large number of sequences in this lineage that are excluded from the strongly supported derived clade? Is it a previously recognised clade that has not been distinguished in this study?

Response: As requested, after adding the outgroup, there were some changes in our phylogenetic tree. We modified it to consist of two groups. The first group comprises three subgroups, mainly distributed in Japan, the Ryukyu Islands, Taiwan, and mainland China. The second group is primarily distributed in Japan, Taiwan, and mainland China.

The genetic distances between lineages I and II are apparently very high for conspecific taxa (>0.10 estimating from Figure 3). It would be desirable to check that the lineages are not cryptic species using a nuclear gene sequence (for example ITS-1 or 2, or a nuclear gene intron) but if this is not possible then parallel series of analyses of population genetic and demographic parameters should be performed for Lineages I and II and any notable differences between the series reported in the text. I note that this has already been done for the AMOVA (two group model).

Response: Our findings are similar to those of Zhao et al. (2017). In Zhao et al.'s study, which primarily focused on the Korean Peninsula to the South China Sea, the maximum genetic distance was approximately 0.09. This result is similar to our study, where our research scope encompassed Taiwan, mainland China, and Japan. Zhao et al. (2017) considered such differences to be within the range of intra-specific variation, and we agree with their interpretation. For Lineages I and II, we conducted population expansion studies and genetic structure analysis, respectively.

The historical demography analyses do not do much to clarify the population history of Monodonta labio in the region with the Bayesian Skyline Plot suggesting a recent size expansion after a previous decline and the Approximate Bayesian Computation suggesting a recent decline after a previous expansion. Size changes might occur if climate changes are combined with barriers to gene during glacial maxima but there have been numerous cycles during the Plio-Pleistocene and it is unclear that the approaches used here can effectively model this. It should be acknowledged in the text that the models used are rather simplistic. Given the disagreement between analyses, causal attributions (e.g. lines 371–373) should be omitted as being merely speculative.

Response: As requested, we revised and deleted it (lines 371–373).

In section 4.3, a date of 0.462 Ma is given for divergence between lineages I and II it is unclear how this is derived (and appears too low if the scale in Figure 3 is correctly labelled).

Response: I apologize for the error in the data. The correct value should be 3.52 million years ago (MA), and we have made the necessary correction.

(line 349) Is there any information as to how long the Kuroshio current has been flowing, and at what strength?

Response: I apologize, but the flow velocity of the Kuroshio varies at different seasons and locations, making it difficult to provide a fixed data. However, it is certain that the direction of the Kuroshio in the waters around Taiwan is from south to north. This directional flow has a significant impact on the migration of marine species.

GenBank accession numbers need to be added.

Response: The GenBank accession numbers are as follows:

Data Availability Statement: All sequences have been deposited in GenBank under the following accession numbers: OQ880826- OQ 880919.

Comments on the Quality of English Language

The writing is generally clear and correct, with occasionally odd word choices that could be revised later.  It is worth noting that the “Ryukyu Islands” are incorrectly referred to as “Ryukyu Island” in the text.  I suggest “investigate” rather than “act” on line 43. The spelling should be “multi-modal” on line 358. “imply that” should be “apply to” or similar (line 356).

Response: As requested, we revised it

Reviewer 2 Report

Population structure and phylogeography of Marine Gastropod

Monodonta labio (Trochidae) along the Northwestern Pacific Coast

Chiu, Bor, Wu, Shieh, and Lin MS

This manuscript seems to be a follow-up on early studies by Donald et al. (2005) and Williams et al. (2010) with more recent studies Monodonta, especially by Yamazaki et al. (2017; 2021 not 2020) and Zhao et al. (2017; 2019). This study extends the genetic comparisons to populations in Taiwan. The results are interesting in that divergent haplotypes called I and II in this manuscript appear to co-occur at some localities (KM, MZ, and PH), and alone on the western (I) and eastern (II) coasts of Taiwan. However, I had to map out the geographic distribution of the haplotypes presented in Figure 3 onto the map of locality codes (Fig. 1) in order to reveal these patterns. This should be done by the authors.

The possibility that there are separate, perhaps cryptic, species was not considered as a possibility by the authors, even though this seems likely. The population-level comparisons seem to be well carried out and this is an interesting extension of the earlier studies but, unfortunately, the authors do not provide enough details to put this comparison into a context of previous studies. This seems important to me given that there are likely at least two cryptic species in M. labio. I think the present study could be worthy of publication but the study as presented here is devoid of a phylogenetic context and without a broader context for a species or species complex reported to be widespread in the Indo-Pacific. As a reader who was not directly familiar with this genus, I had to do my own research to get an idea of how this study relates to the previously mentioned studies, and it is certainly a confusing mess. Perhaps this is why the authors have avoided the confusion of different names used. However, if they do not then this contribution will be difficult to evaluate by readers.  I hope that my comments will help make this a stronger manuscript in the revision that I believe is required.

Consider the portion where M. labio is introduced:

L72: “Marine trochid gastropod Monodonta labio (Linne, 1758) with a wide Indo-Pacific distribution, which is a common gastropod species in the mid intertidal zone belonging to the family (Trochidae), is an herbivorous snail that grazes on microalgae [12].”

This is an awkward sentence, but I see from the literature and from museum records (e.g., USNM) that specimens identified as M. labio from widespread Indo-Pacific localities, apparently occurring as far west and south as Madagascar and many tropical places in the Indian and western Pacific oceans. Here the focus is exclusively on the region between Taiwan and Korea. Already in Donald et al. (2005), it was reported that “M. labio” from Japan was resolved in a different clade than “M. labio” from Australia. Williams et al. (2010; not cited) noted that M. canalifera (Lamark, 1816) appears to be a “cryptic species pair” they label as “A” and “B” but they only included M. labio from Darwin, Australia. Would this currently be considered to be M. australis? I am not sure.

L74: “In East Asia, a total of eleven species and one subspecies of Monodonta have been reported, including two species (M. confuse and M. perplexa perplexa) that undergo a planktonic stage lasting 3 days in their life cycle [13].

Please note the presumed auto-spelling correction typo: M. confusa not confuse. It is normal to give the authority and date as currently recognized in Molluscabase/WoRMS for the first usage of a species name. This is important because there seem to be discrepancies between which species are currently accepted and the species identifications of some sequences in GenBank or other linked databases. It would be helpful if the authors could at least list accession numbers for sequences of M. labio that they do not consider to be that species.

Note also that WoRMS currently accepts only 10 species of this genus worldwide. Perhaps the authors got their tallies partly from Yamazaki et al. (2017):
“The genus Monodonta consists of eleven species and one subspecies, of which eight species and one subspecies are distributed in East Asia: M. australis, M. canalifera, M. confusa, M. glabrata, M. labio, M. viridis, M. neritoides, M. perplexa perplexa, M. perplexa boninensis (listed in Donald et al. 2005).”

There are about 100 other nominal species listed in WoRMS that were originally described within Monodonta that are now assigned to other genera, or are considered junior synonyms of other accepted species, or are of uncertain status, etc. It seems rather important to convince readers that the studied genetically distinct lineages are M. labio and not some other species of the genus. The studies reported here do not place the newly obtained sequences in the context of previous studies, so this makes it very difficult to evaluate how they agree or conflict with previous studies. The focus is exclusively on newly obtained sequences so it is impossible to guess how these distinct haplotypes correspond to what has been demonstrated in previous studies. This is not obvious because previous authors have distinguished haplotype groups within M. labio by letters, not with “I” and “II” as in the present manuscript.

Considering Figure 3, claimed to be “Reconstructed phylogeny of Monodonta labio” I see instead a clustering diagram that is apparently mid-point rooted. This should not be presented as a “phylogenetic estimate of M. labio.” For this it would minimally be necessary to include outgroup species because this is necessary for rooting a phylogenetic estimate. Ideally, one would include the most proximal outgroup species within the genus, perhaps M. confusa + M. australis (see Yamazaki et al. 2017). There are over 2,100 sequences available in GenBank for M. labio and over 3,000 for the genus, Monodonta. Alternatively, they could present this as an unrooted network instead of an arbitrarily rooted “phylogeny.”

Of the M. labio sequences in GenBank there are over 1,000 are COI (cox1) sequences and over 550 are 16S sequences. That is encouraging that there are enough sequences available in GenBank that the authors could place their valuable new geographic sampling of COI and 16S sequences into a broad context for the species and at least show how the newly sampled localities relate to what is known about the phylogeny of M. labio. At one point (L 310), the authors claim to have sampled M. labio more broadly than in previous studies:
“We suggested that we have probably captured the majority of the mtDNA genetic diversity with the larger sample area in M. labio.” It seems odd that they do not at least discuss how their results fit within a broad consideration of all M. labio sequences available within GenBank for this common and widespread species.

Because it was easy for me, I downloaded all available 16S sequences from GenBank and estimated a quick phylogenetic estimate, selecting (arbitrarily) M. perplexa as an outgroup. I will attach the resulting tree along with my review. I did not examine COI sequences. From doing this, I concluded the following:

1) There seem to be two large monophyletic groupings of M. labio, just as in the present manuscript (I & II), but one of them (labeled M. labio-A) had almost all (45 of 46) of the available M. confusa scattered throughout this group (the only exception: “Monodonta_confusa_LC316429_Japan_Osaka_Misaki_1910” was in M. labio-B1). This makes me wonder if the authors have not sequenced two species, M. confusa and M. labio. Again, the authors have given no details for how they decided that all of their specimens were M. labio.

2) The division between M. labio-B1 and M. labio-B2 seems to correspond to Okinawa vs. elsewhere, perhaps similar to the authors’ results, only these are reciprocally monophyletic, not with others paraphyletic to Okinawa as in the authors “phylogeny”. I did not figure out the localities by consulting publications for any of the sequences that had no localities associated with the GenBank records, but I saw at least one sequence for a specimen reported to be from Viet Nam in M. labio-B2. My guess is that almost all of the M. labio 16S sequences in GenBank are from specimens from eastern Asia.

3) M. “australis” was only represented by three specimens from the Ogasawara Islands. This makes me wonder if the population in these remote island has anything at all to do with Lamarck’s species that he reported to have come from Australia.

4) There is one sequence from a specimen reported to be from Darwin, Australia from Donald et al. (2005); as in their early study it grouped with M. canalifera, not with M. labio from Japan (etc.). Perhaps this corresponds to the actual M. labio specimen(s) that Linnaeus described, or perhaps it is a different species. I don’t know.

It is for reasons similar to the analysis that I just completed that it is critically important that the authors refer to specific specimen vouchers that are associated with an appropriate natural history museum. At least they need to have representative specimens that correspond to their different haplotype groups and localities. I found no information included about the specific specimen vouchers included in this study. This needs to be remedied, and these museum numbers should be given in a table (perhaps in a supplement) or associated with the GenBank submission, which apparently had not yet been submitted when the manuscript was submitted: “All sequences have been deposited in GenBank under the following accession numbers:” (none listed).

I see from the “Documented Distribution” in WoRMS that there are multiple accepted species that occur in the vicinity of Taiwan up to Okinawa and to South Korea. It seems that the authors need to address this confusing mess.

The genus Monodonta has been characterized by Herbert (1998):
https://www.biodiversitylibrary.org/page/12441377

The currently accepted species include (with notes and links from WoRMS) at:

https://www.marinespecies.org/aphia.php?p=taxdetails&id=153532

M. labio (Linnaeus, 1758).

I have not found details about any surviving type specimens. I would like to know if any previous authors have possibly restricted the very general (Africa to Asia?) type locality provided by Linnaeus in the linked original description:
https://www.biodiversitylibrary.org/page/34897414

I assume that someone might have figured any type specimen(s) associated with Linnaeus’s description as Trochus labio. This could be important if anyone is going to suggest that M. labia is actually a species complex.

There is an accepted name that seems to have not been used in the literature from where it was described:

Monodonta glabrata Gould, 1861

“Type locality contained in China Sea”

M. glabrata Gould, 1861, is accepted in WoRMS with “type locality contained in China Sea” but there are no sequences identified to this species in GenBank. Literature searches should be completed with “Monodonta glabratum” not “Monodonta glabrata” as currently accepted in WoRMS. Searching at biodiversitylibrary.org I found Gould’s description of it as Monodonta glabratum:
https://www.biodiversitylibrary.org/page/9492434

I also found that there appears to be no type specimen located yet where it is expected at the USNM that matches the original description:

https://www.biodiversitylibrary.org/page/7892534 

It seems that M. glabrata should be considered a nomen inquirendum.

I see that the only Monodonta species included in the second edition of Marine Mollusks in Japan, edited by Takashi Okutani (2017) are:

M. labio labio

M. labio confusa

M. australis

This seems to conflict with recent molecular and morphological structures that recognize M. confusa as a distinct species.

Monodonta confusa Tapparone Canefri, 1874

Type locality Singapore?

https://www.biodiversitylibrary.org/page/11633033

Apparently, according to Yamazaki et al. (2017), M. australis Lamarck, 1822 has been reported only from the Ogasawara Islands of Japan.

Its type locality is “Nouvelle Hollande” = Australia:

https://www.biodiversitylibrary.org/page/13177616

except that WoRMS does not include any details of its distribution within Australia in their documented distribution.

Another nominal species that appears problematic, probably lacking any type material, is:

Monodonta neritoides (R. A. Philippi, 1849)

“accepted > unreplaced junior homonym”

“Distribution Japan to southern China”

The only name that is considered a junior synonym of M. labio in WoRMS is Monodonta trochiformis Grabau & King, 1928, which has a type locality in the Yellow Sea near Beijing. Coan et al. (2015) have included a color image of a syntype of M. trochiformis:

https://doi.org/10.4002/040.058.0206

The authors have not included samples from that far north but it seems likely that this would correspond to their haplotype group “II”.

I am not suggesting that the authors of this manuscript need to revise the taxonomy of M. labio across the Indo-Pacific. That is beyond the scope of this geographically-restricted phylogeographic study. However, I am arguing that they should do more to explain how they identified their material as M. labio and not one of the other co-occurring species of Monodonta in eastern Asia. In particular, I suspect that they might be including what other authors have called M. confusa within M. labio. That would explain what they seem to have at least two species judging from their divergent lineages and the fact that they co-occur at some localities. They also need to relate their haplotype groups to the results from previous studies, and need to discuss the possibility that more than one species might presently be treated under the name, M. labio. They also need to better document the voucher specimens from their study. Someone will eventually need to sort out the taxonomy of M. labio and the authors could certainly improve their manuscript to make it a more valuable contribution for such future studies. I expect that the manuscript could be acceptable for publication if they are able to revise it according to my comments.

Attached: Quick tree of Monodonta 16S available currently in GenBank

The English was generally good. There were only a few scattered places that were in need of fixing, e.g., L316: "are likely to have highly genetic diversity" should be "high".

Author Response

Population structure and phylogeography of Marine Gastropod

Monodonta labio (Trochidae) along the Northwestern Pacific Coast

Chiu, Bor, Wu, Shieh, and Lin MS

This manuscript seems to be a follow-up on early studies by Donald et al. (2005) and Williams et al. (2010) with more recent studies Monodonta, especially by Yamazaki et al. (2017; 2021 not 2020) and Zhao et al. (2017; 2019). This study extends the genetic comparisons to populations in Taiwan. The results are interesting in that divergent haplotypes called I and II in this manuscript appear to co-occur at some localities (KM, MZ, and PH), and alone on the western (I) and eastern (II) coasts of Taiwan. However, I had to map out the geographic distribution of the haplotypes presented in Figure 3 onto the map of locality codes (Fig. 1) in order to reveal these patterns. This should be done by the authors.

 Response:  As requested, we added pie charts depicting the distribution of haplogroups to each population sampling point in Figure 1. “Figure 1. Map showing the 13 sampling localities of Monodonta labio examined in this study. Col-lection sites (circule) correspond to locations given in the text and Table 1. The frequencies of the haplogroups (Figure 3) in each population are displayed on the map. The areas shaded with di-agonal lines in the figure represent the regions that were exposed as sea level during the glacial period, indicating the coastline at that time.”

The possibility that there are separate, perhaps cryptic, species was not considered as a possibility by the authors, even though this seems likely. The population-level comparisons seem to be well carried out and this is an interesting extension of the earlier studies but, unfortunately, the authors do not provide enough details to put this comparison into a context of previous studies. This seems important to me given that there are likely at least two cryptic species in M. labio. I think the present study could be worthy of publication but the study as presented here is devoid of a phylogenetic context and without a broader context for a species or species complex reported to be widespread in the Indo-Pacific. As a reader who was not directly familiar with this genus, I had to do my own research to get an idea of how this study relates to the previously mentioned studies, and it is certainly a confusing mess. Perhaps this is why the authors have avoided the confusion of different names used. However, if they do not then this contribution will be difficult to evaluate by readers.  I hope that my comments will help make this a stronger manuscript in the revision that I believe is required.

 Response:  We are aware that the classification system for this genus has varying perspectives among different scholars. However, our research focuses primarily on the genetic structure of populations and the processes of historical changes. In terms of sequence alignment, our haplogroups are all included in the study published by Zhao et al. in 2017. Therefore, we believe that the populations we are currently studying belong to the species M. labio.

Consider the portion where M. labio is introduced:

L72: “Marine trochid gastropod Monodonta labio (Linne, 1758) with a wide Indo-Pacific distribution, which is a common gastropod species in the mid intertidal zone belonging to the family (Trochidae), is an herbivorous snail that grazes on microalgae [12].”

This is an awkward sentence, but I see from the literature and from museum records (e.g., USNM) that specimens identified as M. labio from widespread Indo-Pacific localities, apparently occurring as far west and south as Madagascar and many tropical places in the Indian and western Pacific oceans. Here the focus is exclusively on the region between Taiwan and Korea. Already in Donald et al. (2005), it was reported that “M. labio” from Japan was resolved in a different clade than “M. labio” from Australia. Williams et al. (2010; not cited) noted that M. canalifera (Lamark, 1816) appears to be a “cryptic species pair” they label as “A” and “B” but they only included M. labio from Darwin, Australia. Would this currently be considered to be M. australis? I am not sure.

 Response:  We revised the sentences. “The marine trochid gastropod Monodonta labio (Linne, 1758) is a common species found in the mid intertidal zone, belonging to the family Trochidae. It exhibits a wide distribution in the Pacific Northwest. This herbivorous snail is known to feed on microalgae [12].”

L74: “In East Asia, a total of eleven species and one subspecies of Monodonta have been reported, including two species (M. confuse and M. perplexa perplexa) that undergo a planktonic stage lasting 3 days in their life cycle [13].

Response:  As requested, we revised it.

Please note the presumed auto-spelling correction typo: M. confusa not confuse. It is normal to give the authority and date as currently recognized in Molluscabase/WoRMS for the first usage of a species name. This is important because there seem to be discrepancies between which species are currently accepted and the species identifications of some sequences in GenBank or other linked databases. It would be helpful if the authors could at least list accession numbers for sequences of M. labio that they do not consider to be that species.

Response: The GenBank accession numbers are as follows:

Data Availability Statement: All sequences have been deposited in GenBank under the following accession numbers: OQ880826- OQ 880919.

Note also that WoRMS currently accepts only 10 species of this genus worldwide. Perhaps the authors got their tallies partly from Yamazaki et al. (2017):
“The genus Monodonta consists of eleven species and one subspecies, of which eight species and one subspecies are distributed in East Asia: M. australisM. canaliferaM. confusaM. glabrataM. labioM. viridisM. neritoidesM. perplexa perplexaM. perplexa boninensis (listed in Donald et al. 2005).”

Response:  As requested, we revised the sentences. “The genus Monodonta consists of eleven species and one subspecies, of which eight species and one subspecies are distributed in East Asia: M. australis, M. canalifera, M. confusa, M. glabrata, M. labio, M. viridis, M. neritoides, M. perplexa perplexa, M. perplexa boninensis (listed in Donald [12]), including two species (M. confusae and M. perplexa perplexa) that undergo a planktonic stage lasting 3 days in their life cycle [13].”

There are about 100 other nominal species listed in WoRMS that were originally described within Monodonta that are now assigned to other genera, or are considered junior synonyms of other accepted species, or are of uncertain status, etc. It seems rather important to convince readers that the studied genetically distinct lineages are M. labio and not some other species of the genus. The studies reported here do not place the newly obtained sequences in the context of previous studies, so this makes it very difficult to evaluate how they agree or conflict with previous studies. The focus is exclusively on newly obtained sequences so it is impossible to guess how these distinct haplotypes correspond to what has been demonstrated in previous studies. This is not obvious because previous authors have distinguished haplotype groups within M. labio by letters, not with “I” and “II” as in the present manuscript.

Response:  We are aware that the classification system for this genus has varying perspectives among different scholars. However, our research focuses primarily on the genetic structure of populations and the processes of historical changes. In terms of sequence alignment, our haplogroups are all included in the study published by Zhao et al. in 2017. Therefore, we believe that the populations we are currently studying belong to the species M. labio.

Considering Figure 3, claimed to be “Reconstructed phylogeny of Monodonta labio” I see instead a clustering diagram that is apparently mid-point rooted. This should not be presented as a “phylogenetic estimate of M. labio.” For this it would minimally be necessary to include outgroup species because this is necessary for rooting a phylogenetic estimate. Ideally, one would include the most proximal outgroup species within the genus, perhaps M. confusa + M. australis (see Yamazaki et al. 2017). There are over 2,100 sequences available in GenBank for M. labio and over 3,000 for the genus, Monodonta. Alternatively, they could present this as an unrooted network instead of an arbitrarily rooted “phylogeny.”

 Response: As requested, we downloaded the relevant sequences from GenBank, including those from Zhao et al.'s study (reference 20 in the paper) and Yamazaki et al.'s study (reference 22). We reconstructed the phylogenetic tree and discussed the relationships between different populations. We revised the sentences. “Other sequences of M. labio, that are available on Genbank (Zhao et al. 2017; Yamazaki et al. 2017; Yamazaki et al. 2021), with additional sequences of M. australis selected as the outgroup.”

Of the M. labio sequences in GenBank there are over 1,000 are COI (cox1) sequences and over 550 are 16S sequences. That is encouraging that there are enough sequences available in GenBank that the authors could place their valuable new geographic sampling of COI and 16S sequences into a broad context for the species and at least show how the newly sampled localities relate to what is known about the phylogeny of M. labio.

Response: As requested, we downloaded the relevant sequences from GenBank, including those from Zhao et al.'s study (reference 20 in the paper) and Yamazaki et al.'s study (reference 22). We reconstructed the phylogenetic tree and discussed the relationships between different populations. We discovered that these sequences have deficiencies in their sampling information from GenBank. As a result, we are unable to use these sequences for analyzing population genetic structure. We can only utilize them for comparative analysis of phylogenetic relationships.

At one point (L 310), the authors claim to have sampled M. labio more broadly than in previous studies:
“We suggested that we have probably captured the majority of the mtDNA genetic diversity with the larger sample area in M. labio.” It seems odd that they do not at least discuss how their results fit within a broad consideration of all M. labio sequences available within GenBank for this common and widespread species.

 Response: As requested, we deleted the sentence.

Because it was easy for me, I downloaded all available 16S sequences from GenBank and estimated a quick phylogenetic estimate, selecting (arbitrarily) M. perplexa as an outgroup. I will attach the resulting tree along with my review. I did not examine COI sequences. From doing this, I concluded the following:

Response: We found sequences on NCBI, and although we downloaded them for analysis, we discovered that these sequences have deficiencies in their sampling information. As a result, we are unable to use these sequences for analyzing population genetic structure. We can only utilize them for comparative analysis of phylogenetic relationships.

1) There seem to be two large monophyletic groupings of M. labio, just as in the present manuscript (I & II), but one of them (labeled M. labio-A) had almost all (45 of 46) of the available M. confusa scattered throughout this group (the only exception: “Monodonta_confusa_LC316429_Japan_Osaka_Misaki_1910” was in M. labio-B1). This makes me wonder if the authors have not sequenced two species, M. confusa and M. labio. Again, the authors have given no details for how they decided that all of their specimens were M. labio.

Response:  We downloaded and analyzed the sequences published by Zhao et al., 2017 and Yamazaki, et al., 2017, 2022. In our lineage â…¡, it included individuals identified as M. confusa by Yamazaki et al. 2017, 2022. However, this cluster also contained individuals identified as M. labio by Zhao et al. 2017. This result indicates that there may still be some controversy regarding the classification of M. confusa and M. labio. Since our study primarily focuses on population genetic structure, we tend to avoid addressing taxonomic issues. Therefore, we agree with Zhao's viewpoint that Lineage â…¡ should still be referred to as M. labio.

2) The division between M. labio-B1 and M. labio-B2 seems to correspond to Okinawa vs. elsewhere, perhaps similar to the authors’ results, only these are reciprocally monophyletic, not with others paraphyletic to Okinawa as in the authors “phylogeny”. I did not figure out the localities by consulting publications for any of the sequences that had no localities associated with the GenBank records, but I saw at least one sequence for a specimen reported to be from Viet Nam in M. labio-B2. My guess is that almost all of the M. labio 16S sequences in GenBank are from specimens from eastern Asia.

Response:  The reviewer's mention of M. labio-B1 group corresponds to the Lineage IA group in my study. This group is predominantly found in the Ryukyu Islands within my sampled population. Similarly, it is also mixed with individuals collected by Zhao et al. in the Gulf of Tonkin, indicating that this lineage is primarily distributed in the Ryukyu Islands, Vietnam, and the Gulf of Tonkin region.

3) M. “australis” was only represented by three specimens from the Ogasawara Islands. This makes me wonder if the population in these remote island has anything at all to do with Lamarck’s species that he reported to have come from Australia.

Response:  M. australis was proposed by Yamazaki et al. in 2017 and 2022. In our phylogenetic tree, this sequence forms a monophyletic group. However, the inter-specific differences are smaller compared to the intra-specific differences observed in our study. Based on the geographic distribution, we consider this species to be valid. However, with the available data, we cannot determine the relationship between this species and the Australian population.

4) There is one sequence from a specimen reported to be from Darwin, Australia from Donald et al. (2005); as in their early study it grouped with M. canalifera, not with M. labio from Japan (etc.). Perhaps this corresponds to the actual M. labio specimen(s) that Linnaeus described, or perhaps it is a different species. I don’t know.

Response: Thank you for this insightful suggestion. However, based on our existing data, we are currently unable to resolve this issue. It involves the species identification done by Donald et al. (2005), and the information available on NCBI is limited. We are unable to determine with certainty which species it corresponds to.

It is for reasons similar to the analysis that I just completed that it is critically important that the authors refer to specific specimen vouchers that are associated with an appropriate natural history museum. At least they need to have representative specimens that correspond to their different haplotype groups and localities. I found no information included about the specific specimen vouchers included in this study. This needs to be remedied, and these museum numbers should be given in a table (perhaps in a supplement) or associated with the GenBank submission, which apparently had not yet been submitted when the manuscript was submitted: “All sequences have been deposited in GenBank under the following accession numbers:” (none listed).

I see from the “Documented Distribution” in WoRMS that there are multiple accepted species that occur in the vicinity of Taiwan up to Okinawa and to South Korea. It seems that the authors need to address this confusing mess.

The genus Monodonta has been characterized by Herbert (1998):
https://www.biodiversitylibrary.org/page/12441377

The currently accepted species include (with notes and links from WoRMS) at:

https://www.marinespecies.org/aphia.php?p=taxdetails&id=153532

  1. labio (Linnaeus, 1758).

I have not found details about any surviving type specimens. I would like to know if any previous authors have possibly restricted the very general (Africa to Asia?) type locality provided by Linnaeus in the linked original description:
https://www.biodiversitylibrary.org/page/34897414

I assume that someone might have figured any type specimen(s) associated with Linnaeus’s description as Trochus labio. This could be important if anyone is going to suggest that M. labia is actually a species complex.

There is an accepted name that seems to have not been used in the literature from where it was described:

Monodonta glabrata Gould, 1861

“Type locality contained in China Sea”

  1. glabrata Gould, 1861, is accepted in WoRMS with “type locality contained in China Sea” but there are no sequences identified to this species in GenBank. Literature searches should be completed with “Monodonta glabratum” not “Monodonta glabrata” as currently accepted in WoRMS. Searching at biodiversitylibrary.org I found Gould’s description of it as Monodonta glabratum:
    https://www.biodiversitylibrary.org/page/9492434

I also found that there appears to be no type specimen located yet where it is expected at the USNM that matches the original description:

https://www.biodiversitylibrary.org/page/7892534 

It seems that M. glabrata should be considered a nomen inquirendum.

I see that the only Monodonta species included in the second edition of Marine Mollusks in Japan, edited by Takashi Okutani (2017) are:

  1. labio labio
  2. labio confusa
  3. australis

This seems to conflict with recent molecular and morphological structures that recognize M. confusa as a distinct species.

Monodonta confusa Tapparone Canefri, 1874

Type locality Singapore?

https://www.biodiversitylibrary.org/page/11633033

Apparently, according to Yamazaki et al. (2017), M. australis Lamarck, 1822 has been reported only from the Ogasawara Islands of Japan.

Its type locality is “Nouvelle Hollande” = Australia:

https://www.biodiversitylibrary.org/page/13177616

except that WoRMS does not include any details of its distribution within Australia in their documented distribution.

Another nominal species that appears problematic, probably lacking any type material, is:

Monodonta neritoides (R. A. Philippi, 1849)

“accepted > unreplaced junior homonym”

“Distribution Japan to southern China”

The only name that is considered a junior synonym of M. labio in WoRMS is Monodonta trochiformis Grabau & King, 1928, which has a type locality in the Yellow Sea near Beijing. Coan et al. (2015) have included a color image of a syntype of M. trochiformis:

https://doi.org/10.4002/040.058.0206

The authors have not included samples from that far north but it seems likely that this would correspond to their haplotype group “II”.

I am not suggesting that the authors of this manuscript need to revise the taxonomy of M. labio across the Indo-Pacific. That is beyond the scope of this geographically-restricted phylogeographic study. However, I am arguing that they should do more to explain how they identified their material as M. labio and not one of the other co-occurring species of Monodonta in eastern Asia. In particular, I suspect that they might be including what other authors have called M. confusa within M. labio. That would explain what they seem to have at least two species judging from their divergent lineages and the fact that they co-occur at some localities. They also need to relate their haplotype groups to the results from previous studies, and need to discuss the possibility that more than one species might presently be treated under the name, M. labio. They also need to better document the voucher specimens from their study. Someone will eventually need to sort out the taxonomy of M. labio and the authors could certainly improve their manuscript to make it a more valuable contribution for such future studies. I expect that the manuscript could be acceptable for publication if they are able to revise it according to my comments.

Response: We downloaded and analyzed the sequences published by Zhao et al., 2017 and Yamazaki, et al., 2017, 2022. In our lineage â…¡, it included individuals identified as M. confusa by Yamazaki et al. 2017, 2022. However, this cluster also contained individuals identified as M. labio by Zhao et al. 2017. This result indicates that there may still be some controversy regarding the classification of M. confusa and M. labio. Since our study primarily focuses on population genetic structure, we tend to avoid addressing taxonomic issues. Therefore, we agree with Zhao's viewpoint that Lineage â…¡ should still be referred to as M. labio.

In the supplementary table, we provide detailed information regarding the sampling GPS coordinates of each individual and their corresponding sequence identifiers on NCBI. Although our study primarily focuses on population genetics and population dynamics, these sequences can be utilized for future studies on the distribution and phylogenetic relationships of different species within this genus.

Attached: Quick tree of Monodonta 16S available currently in GenBank

Reviewer 3 Report

Thanks for the opportunity for reviewing this manuscript. Generally the paper is well written and the results support the conclusion. I suggest publish this manuscript after some minor edition. However, the genetic distance between lineages I and II are enormous. It looks more like two different species instead of one. Another possibility is that the authors might have sequenced the pseudogene. The authors should carefully check the morphological identification and the sequences of their specimen. If the specimens are incorrectly identified or pseudogene was used in the analysis, the manuscript shouldn’t be published until all confounding data been removed from the analysis.

1.       I suggest that the authors should provide some extract information in the supplementary materials: 1) as the manuscript included a hypothesis regarding the glaciation separation of populations, a map showing what the studied area looked like during glacial period can be added to allow better understanding of the story. 2) Please provide figures of detailed NJ, ML and BI trees in the supplementary materials. Especially the ML and BI trees are more reliable compared to NJ tree, in terms of phylogenetic analysis. 3) Details of each sample used in this study, including GPS coordinates and GenBank accession numbers. If the specimen records have been updated to BOLD, please provide BOLD submission ID.

2.       There is another paper that the authors should cite and used in their dataset: Yamazaki, D., Miura, O., Ikeda, M. et al. Genetic diversification of intertidal gastropoda in an archipelago: the effects of islands, oceanic currents, and ecology. Mar Biol 164, 184 (2017). https://doi.org/10.1007/s00227-017-3207-9. This paper used the same gene regions. The authors should use the sequence data published in Yamazaki’s research together in this manuscript.

3.       Another critical point is the difference between lineage I and II. Based on Figure 4, these two lineages have more than 100 bp differences that is 10% of genetic distance in COI + 16s. This make me think lineage I and II are different species or the authors had obtained the pseudogene which is common in COI gene analysi. I urge the authors double check their species identification and their sequences. If the species identification is not correct or pseudogene sequences presented, this manuscript shouldn’t be published until it has been fixed.

4.       Based on Figure 3. Lineage IB shouldn’t be considered as a lineage. A lineage is a group of specimens diverged from a common ancestor. Based on the tree IB is a polyphyletic group. In addition, the branch supports of within this polyphyletic group are not shown. It is not possible to tell how well is the grouping. However, based on Figure 4, IA and IB are genetically distinct but based on Figure 4 there are two groups of samples in lineage II as well. I am not sure why the authors didn’t consider that.

5.       Figure 4 is very difficult to see details such as bars across the branches. Also the authors didn’t indicated which software they used to create this network analysis.

Author Response

Thanks for the opportunity for reviewing this manuscript. Generally the paper is well written and the results support the conclusion. I suggest publish this manuscript after some minor edition. However, the genetic distance between lineages I and II are enormous. It looks more like two different species instead of one.

Response: Our findings are similar to those of Zhao et al. (2017). In Zhao et al.'s study, which primarily focused on the Korean Peninsula to the South China Sea, the maximum genetic distance was approximately 0.09. This result is similar to our study, where our research scope encompassed Taiwan, mainland China, and Japan. Zhao et al. (2017) considered such differences to be within the range of intra-specific variation, and we agree with their interpretation. For Lineages I and II, we conducted population expansion studies and genetic structure analysis, respectively.

Another possibility is that the authors might have sequenced the pseudogene.

Response: We used the COI and 16S genes as genetic markers, which are commonly used in studies of this species. For example, in Zhao et al. (citation 20 in the article) and Yamazaki et al. (citation 22).

The authors should carefully check the morphological identification and the sequences of their specimen. If the specimens are incorrectly identified or pseudogene was used in the analysis, the manuscript shouldn’t be published until all confounding data been removed from the analysis.

Response: We have been very rigorous in specimen identification. After obtaining the sequences, we performed a BLAST comparison on NCBI to confirm the correct species before proceeding with the analysis. We also downloaded sequences from previous studies and included them in our analysis to ensure that our samples are represented in other studies of the same species. Our sequences do not contain any sequences from different species mixed in.

  1. I suggest that the authors should provide some extract information in the supplementary materials: 1) as the manuscript included a hypothesis regarding the glaciation separation of populations, a map showing what the studied area looked like during glacial period can be added to allow better understanding of the story.

Response: As requested, we added the coastline during the glacial period in Figure 1. “Figure 1. Map showing the 13 sampling localities of Monodonta labio examined in this study. Col-lection sites (circule) correspond to locations given in the text and Table 1. The frequencies of the haplogroups (Figure 3) in each population are displayed on the map. The areas shaded with di-agonal lines in the figure represent the regions that were exposed as sea level during the glacial period, indicating the coastline at that time.”

2) Please provide figures of detailed NJ, ML and BI trees in the supplementary materials. Especially the ML and BI trees are more reliable compared to NJ tree, in terms of phylogenetic analysis.

Response: As requested, we added ML and BI trees in the supplementary materials.

 3) Details of each sample used in this study, including GPS coordinates and GenBank accession numbers. If the specimen records have been updated to BOLD, please provide BOLD submission ID.

Response: As requested, we revised it. “In Table 1, we listed the latitude and longitude coordinates of the sampling points for each population.”

  1. There is another paper that the authors should cite and used in their dataset: Yamazaki, D., Miura, O., Ikeda, M. et al.Genetic diversification of intertidal gastropoda in an archipelago: the effects of islands, oceanic currents, and ecology. Mar Biol 164, 184 (2017). https://doi.org/10.1007/s00227-017-3207-9. This paper used the same gene regions. The authors should use the sequence data published in Yamazaki’s research together in this manuscript.

Response: As requested, we downloaded the relevant sequences from GenBank, including those from Zhao et al.'s study (reference 20 in the paper) and Yamazaki et al.'s study (reference 22). We reconstructed the phylogenetic tree and discussed the relationships between different populations.

  1. Another critical point is the difference between lineage I and II. Based on Figure 4, these two lineages have more than 100 bp differences that is 10% of genetic distance in COI + 16s. This make me think lineage I and II are different species or the authors had obtained the pseudogene which is common in COI gene analysi. I urge the authors double check their species identification and their sequences. If the species identification is not correct or pseudogene sequences presented, this manuscript shouldn’t be published until it has been fixed.

Response: Our findings are similar to those of Zhao et al. (2017). In Zhao et al.'s study, which primarily focused on the Korean Peninsula to the South China Sea, the maximum genetic distance was approximately 0.09. This result is similar to our study, where our research scope encompassed Taiwan, mainland China, and Japan. Zhao et al. (2017) considered such differences to be within the range of intra-specific variation, and we agree with their interpretation. For Lineages I and II, we conducted population expansion studies and genetic structure analysis, respectively.

  1. Based on Figure 3. Lineage IB shouldn’t be considered as a lineage. A lineage is a group of specimens diverged from a common ancestor. Based on the tree IB is a polyphyletic group. In addition, the branch supports of within this polyphyletic group are not shown. It is not possible to tell how well is the grouping. However, based on Figure 4, IA and IB are genetically distinct but based on Figure 4 there are two groups of samples in lineage II as well. I am not sure why the authors didn’t consider that.

Response: As requested, after adding the outgroup, there were some changes in our phylogenetic tree. We modified it to consist of two groups. The first group comprises three subgroups, mainly distributed in Japan, the Ryukyu Islands, Taiwan, and mainland China. The second group is primarily distributed in Japan, Taiwan, and mainland China.

  1. Figure 4 is very difficult to see details such as bars across the branches. Also the authors didn’t indicated which software they used to create this network analysis.

Response: Haplotype networks were constructed using the MINSPNET algorithm implemented in Popart [35].

Round 2

Reviewer 1 Report

The authors are to be commended for including the relevant GenBank sequences in the study, which has resulted in a more detailed phylogeographic study that can be placed in the context of previous work.  Additional text has been added regarding the status of the lineages (are they subspecies of a single species or do they represent distinct specie). The authors continue to regard them as conspecific. This is mentioned in the Discussion but should also be stated in the Introduction. I also think this choice either needs to be better justified or a statement placed somewhere in the Discussion that the question remains open and that re-analysis and re-appraisal would be required if the lineages were shown to represent distinct species.

As I noted in my initial review, the historical demography analyses do not do much to clarify the population history of Monodonta labio in the region as they give contradictory results. I have suggested a point in the text (see the attached file) where a general caveat about the application of simple models such as those used here is unlikely to provide accurate phylogeographic histories where the evolution of a species has been influenced by multiple glacial cycles.

I have made a number of suggested linguistic changes on the attached file. Not all previous changes have been completed by removal of the original text after the insertion of new material. This needs to be checked throughout the manuscript.

Author Response

Comments and Suggestions for Authors

The authors are to be commended for including the relevant GenBank sequences in the study, which has resulted in a more detailed phylogeographic study that can be placed in the context of previous work.  Additional text has been added regarding the status of the lineages (are they subspecies of a single species or do they represent distinct specie). The authors continue to regard them as conspecific. This is mentioned in the Discussion but should also be stated in the Introduction. I also think this choice either needs to be better justified or a statement placed somewhere in the Discussion that the question remains open and that re-analysis and re-appraisal would be required if the lineages were shown to represent distinct species.

Response: We acknowledge the feedback from other reviewers and agree that the two main lineages we currently observe belong to two distinct species, namely Monodonta labio and M. confusa. As a result, we have made significant changes in our analysis, leading to a complete reevaluation and reanalysis of our findings.

As I noted in my initial review, the historical demography analyses do not do much to clarify the population history of Monodonta labio in the region as they give contradictory results. I have suggested a point in the text (see the attached file) where a general caveat about the application of simple models such as those used here is unlikely to provide accurate phylogeographic histories where the evolution of a species has been influenced by multiple glacial cycles.

Response: We are aware that analyses regarding population history often present conflicting results. Therefore, we attempted to utilize the ABC model to elucidate the population history. Naturally, we fully agree that the impact of past complex historical events on populations is by no means singular. Thank you for the constructive feedback from the reviewers. We have made the necessary revisions based on their suggestions.

Response: As requested, we revised it.

Reviewer 2 Report

    The authors have substantially improved some aspects of their manuscript. The addition of pie charts to their map figure helps the reader visualize the distribution of haplotypes.

   It is good to hear they have now downloaded and analyzed other available relevant sequences available in GenBank. The outgroup-rooted phylogenetic estimate is a substantial improvement. However, for reasons I explain below, the authors still need to add other proximal outgroup Monodonta species because the single one they chose from the remote Ogasawara Islands happens to have been re-identified as M. confusa (based on DNA sequence results alone) by Yamazaki et al. (2017). Because the authors have declared it as the outgroup, this possibility that it could be part of the ingroup is not allowed. I am sorry if my 16S tree accompanying my earlier review implied otherwise, but this is still also a possible result.

    The authors have now listed pending (unreleased) GenBank accession numbers but questions remain about how the present data relates to other studies. The responses to reviewers might make it possible to match the authors' modified assignment to lineages "I" and "II" to previous studies but I am not finding any mention in their manuscript revision of how they correspond, and this is unacceptable. Zhao et al. (2017), whose species treatment the authors claim to follow, refer to "clades" A-E. Yamazaki et al. (2017) refer to groupings "A to C" within M. labio and groupings "D to F" within M. confusa. I appreciated the explanations of how the haplotype groups correspond in their reviewer responses but it needs to be clearly summarized in the manuscript, especially because the authors are using their own separate numbering system.

    Interestingly, as pointed out in my first review, Yamazaki et al. (2017) reassign the two sequences of M. "australis" from the Ogasawara Islands as part of their grouping "D" of M. confusa. It is clear from their labeling that they have re-identified the specimens from these sequences, even if they do this only with DNA. They could justify this re-identification because of broader outgroup comparisons they made within Monodonta, including M. "labio" in GenBank from Australia and "M. australis" from East Africa. In their results, these are not the same as "M. australis" from the Ogasawara Islands. In fact, other Monodonta species are closer to what they call M. labio and M. confusa. There is still a need to sample more broadly but because it is known that the type locality of M. australis is Australia, there seems no valid reason to reject Yamazaki et al.'s re-identification of the sequenced specimens of M. "australis" from the Ogasawara Islands as M. confusa.

    My inclusion of a 16S-only quick neighbor joining tree with my first review, which depicts M. "australis" from the Ogasawara Islands as sister lineage of all M. labio, or M. labio + M. confusa, was probably misleading. It seems to have led the authors to their choice to ignore more distal outgroups and this is critically important because the present authors have declared M. "australis" from the Ogasawara Islands as their outgroup, in contrast to the published findings of Yamazaki et al. (2017).

I apologize that the following is technically dense. It is quite possible that a more rigorous analysis including node support estimates could reveal that the declared outgroup where they have chosen to root their tree, could be a member of the ingroup. For example, this is the case for the topology reported by Yamazaki et al. (2017). In fact, they found that M. "australis" (Ogasawara Islands) is supported within their M. confusa grouping 83% bootstrap support, part of their "M. confusa group" in a separate figure. Their result is the most complete one published to date, which means the present phylogenetic estimate is inappropriately rooted. This would also affect the claimed monophyletic grouping of their modified Lineages "I" and "II" in the authors' response to reviewers. However, if they had included other appropriately proximal outgroups, as was my suggestion, either the topology result from Yamazaki et al. or the topology from the 16S tree I sent could support the claim of monophyletic lineages of the present "I" and "II" but if support is for the Yamazaki et al. result, then M. "australis" from Ogasawara Islands should be added to Lineage "II" (= M. confusa in Yamazaki et al. 2017). Or, Monodonta sp. from Ogasawara Islands might be resolved as sister lineage to the two combined lineages "I" and "II" as presently defined by the authors. With either definition of Lineage "II," both lineages "I" and "II" could be monophyletic. But this will require adding more outgroups to resolve the position of the declared outgroup.

    The authors give no details about the fate of any specimens used in the study. They are apparently not saved so are unavailable for any future studies for other purposes such as morphological or additional molecular study, which seems a shame.

    The authors argue that the one- vs. two-species controversy in unimportant to their manuscript because "our research focuses primarily on the genetic structure of populations and the processes of historical changes." If there are two species, by a variety of species concepts, this directly impacts any division of populations in the analysis. Comparing populations of different species is an interspecific comparison. The fact that lineages "I" and "II" co-occur within a portion of the study area would seem to put the burden of proof on the authors to demonstrate there is a single species with genetic connectivity. There are nuclear-based methods for testing such reproductive compatibility.

    The authors dismiss the point made by all three reviewers who all pointed out that there appear to be at least two separate species within what the authors are asserting is one, Monodonta labio. Their  particular response:

    "Response: We are aware that the classification system for this genus has varying perspectives among different scholars. However, our research focuses primarily on the genetic structure of populations and the processes of historical changes. In terms of sequence alignment, our haplogroups are all included in the study published by Zhao et al. in 2017. Therefore, we believe that the populations we are currently studying belong to the species M. labio."

    Here is the history that Zhao et al. (2017) covered that is never explained in the present manuscript:

    "Within this species, two commonly recognized Japanese subspecies, referred to as Monodonta labio labio and Monodonta labio confusa, are documented based on differences in shell morphology (Higo, Callomon, & GotoÌ„, 1999). Donald, Kennedy, and Spencer (2005) found that they are genetically divergent and treated them as separate species, Monodanta labio and Monodonta confusa. In addition, they noticed that individuals obtained from Australia turned out to be genetically distinct from the two Japanese subspecies."

    In the same year, in contrast to the single treatment as a single species, Yamazaki et al. (2017) concluded with more complete sampling of the genus across the Indo-Pacific:

    "This study shows that the genus Monodonta in the Indo-West Pacific was genetically divided into three major clades: the East African clade, Asia-Australian clade, and East Asian clade (Fig. 2). These results imply that M. labio and M. australis, which have been thought of as widely distributed, are not single species."

    Their East Asian clade consists of not only a "M. labio group" and "M. confusa group" but also M. neritoides and "M. perplexa group".

    Moreover, they documented  a contrasting ecological difference in their 2021 paper:

    "We found that M. labio was dominant in sheltered habitats and M. perplexa was dominant in wave-exposed habitats, while M. confusa showed no habitat specificity. This indicates that M. labio and M. perplexa are habitat specialists regarding wave exposure, while M. confusa is a generalist."

    In the previous review, it was pointed out that other sequences in GenBank are assigned to M. confusa, a species that is accepted in WoRMS. The authors cannot simply assert that M. labio is one species because Zhao et al. (2017) said so. When there is a controversy, the authors need to address this information near the beginning of their introduction to this taxon. I learned quite a bit in the responses from the authors that I wished was included in either version of the authors' manuscript. Now there are a couple of sentences near the end (!) of the manuscript:

    We incorporated sequences published by other studies into

    our analysis and found that they can be divided into two groups, lineage I and II. Among

    the individuals in the lineage II group, Zhao et al. [20] identified them as M. labio, while

    Yamazaki et al. [20, 21] identified them as M. confusa. Due to the wide distribution range

    of this group, in our study, to avoid confusion in the classification system and to focus on

    the population's historical dynamics, we agree with the viewpoint of Zhao et al. [20] and

    consider the individuals in lineage II as M. labio (Figure S1, Table S2).

    Please, do not make a reader find out about that M. labio is being treated more broadly than is currently accepted in WoRMS. The authors still treat it as a single species for the entire manuscript despite apparent genetic evidence to the contrary that all three reviewers noticed. It seems that the case could be easily made for even more than two species. An appropriate place to bring up details about the accepted taxonomy in WoRMs, as earlier advocated by Yamazaki et al. (cited as 21), is in line 92, where the authors misrepresent their conclusion; they imply that Yamazaki and co-authors concluded there are two lineages within M. labio:

" Another similar study proposed that M. labio was primarily separated into two lineages with a boundary between the Japanese mainland and the Ryukyu Islands [21]. The genetic structure of M. labio is also influenced by ecological factors such as habitat specificity and adaptability. Molecular studies have suggested lower genetic diversity and greater genetic differentiation in M. labio, possibly due to these ecological factors [22]."

    By not even explaining that there is a controversy that frankly seems to be favoring the two species hypothesis, judging from the acceptance of M. confusa in WoRMS, the authors appear to be overly stubborn to not explain and justify, in the Introduction, their decision to accept M. confusa as a synonym of M. labio, contrary to its treatment in WoRMS. By waiting until the last paragraph of their manuscript to bring up the current controversy in the literature, and never mentioning the accepted species ironically named, M. confusa, they have contributed to "confusion in the classification system." This is unacceptable.

    However, I would personally have no problem if the authors could, in a more substantial revision, sufficiently justify their choice of the broader species concept used by Zhao et al. (2017), not by assertion but based on evidence from their results. For example, they could further cite the morphometric conclusions in the later paper by Zhao et al. (2019), which at first glance to me seems to show substantial overlap in shell shape at least between the Zhao et al. (2017) "clades." There is also the issue that the analysis of Yamazaki et al. (2017) does not firmly support (bootstrap support is < 50%) the monophyly of their restricted M. labio, whereas, their M. confusa (with M. "australis" - Ogasawara) has 83% bootstrap support. From the haplotype network figure  with redrawn Lineage "I" vs. "II" boundaries, it is possible that "I-C" sometimes groups with "I-B" + "I-A" and sometimes groups with "II" and this alternative topology needs to be considered. This analysis has added more geographic sampling that could be relevant to resolving these basal branching patterns of the "I" + "II" grouping, with or without the Ogasawara sequences included within "II."

Here are more technical suggestions of how a long-branch part of the ingroup taxon can "bounce around" in the tree and reduce resolution compared to if it is trimmed from the analysis. If combined phylogenetic estimates support the Ogasawara branch as part of the ingroup, it is still likely to be a long branch. The inclusion of a highly divergent ingroup sequence can obscure resolution of branching pattern, so I would recommend exploring deleting or restoring the Ogasawara samples and report on the impact on the apparent "I" vs. "II" basal split. Overall, the authors could find in a combined 16S+cox1 result that there is no clear evidence for splitting these lineages into only two species, instead supporting an unresolved polytomy of multiple lineages. The latter finding might support a broad M. labio species concept. But they need to include more proximal outgroups to evaluate this possibility. I recommend they include outgroups that arise from two nodes "below" the ingroup, which might or might not include M. sp. from the Ogasawara Islands as the most proximal outgroup. That is a result of the analysis, not something that is declared.

    I felt like I had to dig out important details on previous molecular systematics and biogeography research on Monodonta that I should have been introduced to early on, along with a generally good description of the fluctuating land barriers to dispersal during Pleistocene glacial intervals. The authors have made their lack of interest in the one- vs. two-species controversy clear, and I understand that there could be a case for a broad definition of a species. But in this case the species controversies happen to be highly relevant to their genetic study. The controversy cannot simply be ignored until the last paragraph. Some of the present results could be open to alternative interpretations if more than one species is represented in their samples, or at least the reader deserves to learn about them much earlier.

     As the authors point out, this is one of the most common snails on the shores of the study area. Previous authors, from studies dating to 2005 but most recently by Yamazaki et al. (2017; 2021), have at least convinced the knowledgeable editors at WoRMS (MolluscaBase) that both M. labio and M. confusa are currently accepted. Can the authors cite a study more recent than 2017 that has disputed the two-species interpretation? It is even more interesting that habitat differences exist between species or intraspecific lineages where they co-occur (Yamazaki et al. 2021). The possibility at least needs to be addressed as it relates to results of this study.

    With respect to the ABC software employed in this study, I am not directly familiar with the use of this particular software but I do think it must matter substantially if one chooses to group two species as if this was a priori a single group. It could easily inflate estimates such as effective population size if there is no gene flow between them. If there is previous evidence in the literature that substantial hybridization between M. labio and M. confusa, as accepted currently in WoRMS, then the situation could be different.

    It did not help that the authors have redundantly posted the same flawed assertion response of citing only a single study to justify their taxonomy to multiple objections raised in their response to reviewers. Because mitochondrial markers are maternally inherited with no opportunity to directly test for genetic connectivity, they cannot assume a potential for interbreeding across their "populations," and having Lineage "I" and "II" co-occurring in part of their study area might itself suggest that at least two species are present.

   I have reached the conclusion that only another major revision would potentially make this an acceptable manuscript for publication. Their present revision is minimal and does not sufficiently address the criticisms made by all three reviewers. I am close to recommending the manuscript could be rejected if for no other reason that the authors seem think that historical cladogenetic events (i.e., speciation) are not relevant to their analysis of "populations within a single species" and "the processes of historical changes." Evolution is not only anagenesis (drift and selection), it also involves cladogenesis. When framed in a biogeographic context, this makes an important difference the authors need to address. On the other hand, I still think a more substantial manuscript revision than the authors have provided could be a valuable contribution to the literature on these common and interesting snails that have relatively brief planktonic larval duration.

Specific comments:

Line 50, is "repressively" the right word?

Lines 72-79, I have no idea of the authors final text from, and at some other places where there are edits. Guessing at how fragments of old versions will be eliminated still leaves sentences that are convoluted and use the wrong words. I need to be able to read entire sentences. It is partly because of the confusing way this manuscript has been edited to leave in old text.

Line 85, "In contrast,…"

Line 105, suggest "mainland China, Jeju Island in Korea [was missing], and representing the Ryukyu Islands and more northern localities within Japan."

Or: "extending comparisons from previous studies to both coasts of Taiwan, expanding sampling from mainland China, Jeju Island in Korea, the Ryukyu Islands and more northern localities within Japan."

Also note that "Ryukyu" as a header in Table 1 needs to be fixed. Plus, it is part of Japan.

Line 106, suggest "simulations, which offer…"

Line 168-169: The manuscript in general is insufficiently proofread.

I did not make further specific comments.

See comments for lines 72-79. There are similar other problems.

Author Response

Comments and Suggestions for Authors

    The authors have substantially improved some aspects of their manuscript. The addition of pie charts to their map figure helps the reader visualize the distribution of haplotypes.

   It is good to hear they have now downloaded and analyzed other available relevant sequences available in GenBank. The outgroup-rooted phylogenetic estimate is a substantial improvement. However, for reasons I explain below, the authors still need to add other proximal outgroup Monodonta species because the single one they chose from the remote Ogasawara Islands happens to have been re-identified as M. confusa (based on DNA sequence results alone) by Yamazaki et al. (2017). Because the authors have declared it as the outgroup, this possibility that it could be part of the ingroup is not allowed. I am sorry if my 16S tree accompanying my earlier review implied otherwise, but this is still also a possible result.

Response: We revised the choice of outgroup and selected M. perplexa as the outgroup species.

     The authors have now listed pending (unreleased) GenBank accession numbers but questions remain about how the present data relates to other studies. The responses to reviewers might make it possible to match the authors' modified assignment to lineages "I" and "II" to previous studies but I am not finding any mention in their manuscript revision of how they correspond, and this is unacceptable. Zhao et al. (2017), whose species treatment the authors claim to follow, refer to "clades" A-E. Yamazaki et al. (2017) refer to groupings "A to C" within M. labio and groupings "D to F" within M. confusa. I appreciated the explanations of how the haplotype groups correspond in their reviewer responses but it needs to be clearly summarized in the manuscript, especially because the authors are using their own separate numbering system.

Response: As requested, we revised it.  We have redefined the species relationship of lineages "I" and "II," and in the discussion, we highlight the relationship between our study and the previously mentioned study's lineages.

     Interestingly, as pointed out in my first review, Yamazaki et al. (2017) reassign the two sequences of M. "australis" from the Ogasawara Islands as part of their grouping "D" of M. confusa. It is clear from their labeling that they have re-identified the specimens from these sequences, even if they do this only with DNA. They could justify this re-identification because of broader outgroup comparisons they made within Monodonta, including M. "labio" in GenBank from Australia and "M. australis" from East Africa. In their results, these are not the same as "M. australis" from the Ogasawara Islands. In fact, other Monodonta species are closer to what they call M. labio and M. confusa. There is still a need to sample more broadly but because it is known that the type locality of M. australis is Australia, there seems no valid reason to reject Yamazaki et al.'s re-identification of the sequenced specimens of M. "australis" from the Ogasawara Islands as M. confusa. My inclusion of a 16S-only quick neighbor joining tree with my first review, which depicts M. "australis" from the Ogasawara Islands as sister lineage of all M. labio, or M. labio + M. confusa, was probably misleading. It seems to have led the authors to their choice to ignore more distal outgroups and this is critically important because the present authors have declared M. "australis" from the Ogasawara Islands as their outgroup, in contrast to the published findings of Yamazaki et al. (2017).

Response:  As questioned by the reviewers, our original outgroup was M. australis. To address this dispute, we have now changed the outgroup to M. perplexa in order to avoid controversy.

I apologize that the following is technically dense. It is quite possible that a more rigorous analysis including node support estimates could reveal that the declared outgroup where they have chosen to root their tree, could be a member of the ingroup. For example, this is the case for the topology reported by Yamazaki et al. (2017). In fact, they found that M. "australis" (Ogasawara Islands) is supported within their M. confusa grouping 83% bootstrap support, part of their "M. confusa group" in a separate figure. Their result is the most complete one published to date, which means the present phylogenetic estimate is inappropriately rooted. This would also affect the claimed monophyletic grouping of their modified Lineages "I" and "II" in the authors' response to reviewers. However, if they had included other appropriately proximal outgroups, as was my suggestion, either the topology result from Yamazaki et al. or the topology from the 16S tree I sent could support the claim of monophyletic lineages of the present "I" and "II" but if support is for the Yamazaki et al. result, then M. "australis" from Ogasawara Islands should be added to Lineage "II" (= M. confusa in Yamazaki et al. 2017). Or, Monodonta sp. from Ogasawara Islands might be resolved as sister lineage to the two combined lineages "I" and "II" as presently defined by the authors. With either definition of Lineage "II," both lineages "I" and "II" could be monophyletic. But this will require adding more outgroups to resolve the position of the declared outgroup.

Response:  As questioned by the reviewers, our original outgroup was M. australis. To address this dispute, we have now changed the outgroup to M. perplexa in order to avoid controversy. We appreciate the suggestions from the reviewers. By changing the outgroup species, the issue of phylogenetic relationships has been resolved.

    The authors give no details about the fate of any specimens used in the study. They are apparently not saved so are unavailable for any future studies for other purposes such as morphological or additional molecular study, which seems a shame.

Response:  Regarding the sample issue, we also apologize. Due to the lack of attention to subtle differences in past morphological classifications, during the study process, it was necessary to break the shell to extract sufficient tissue for DNA extraction. As samples were not preserved, we deeply regret this situation.

    The authors argue that the one- vs. two-species controversy in unimportant to their manuscript because "our research focuses primarily on the genetic structure of populations and the processes of historical changes." If there are two species, by a variety of species concepts, this directly impacts any division of populations in the analysis. Comparing populations of different species is an interspecific comparison. The fact that lineages "I" and "II" co-occur within a portion of the study area would seem to put the burden of proof on the authors to demonstrate there is a single species with genetic connectivity. There are nuclear-based methods for testing such reproductive compatibility.

Response:  Due to the limitations of genetic markers, mitochondrial DNA is maternally inherited and cannot detect issues related to hybridization. Therefore, there are inherent constraints in species delineation and studies of gene flow.

     The authors dismiss the point made by all three reviewers who all pointed out that there appear to be at least two separate species within what the authors are asserting is one, Monodonta labio. Their  particular response:

    "Response: We are aware that the classification system for this genus has varying perspectives among different scholars. However, our research focuses primarily on the genetic structure of populations and the processes of historical changes. In terms of sequence alignment, our haplogroups are all included in the study published by Zhao et al. in 2017. Therefore, we believe that the populations we are currently studying belong to the species M. labio."

    Here is the history that Zhao et al. (2017) covered that is never explained in the present manuscript: "Within this species, two commonly recognized Japanese subspecies, referred to as Monodonta labio labio and Monodonta labio confusa, are documented based on differences in shell morphology (Higo, Callomon, & GotoÌ„, 1999). Donald, Kennedy, and Spencer (2005) found that they are genetically divergent and treated them as separate species, Monodanta labio and Monodonta confusa. In addition, they noticed that individuals obtained from Australia turned out to be genetically distinct from the two Japanese subspecies."

    In the same year, in contrast to the single treatment as a single species, Yamazaki et al. (2017) concluded with more complete sampling of the genus across the Indo-Pacific: "This study shows that the genus Monodonta in the Indo-West Pacific was genetically divided into three major clades: the East African clade, Asia-Australian clade, and East Asian clade (Fig. 2). These results imply that M. labio and M. australis, which have been thought of as widely distributed, are not single species."

    Their East Asian clade consists of not only a "M. labio group" and "M. confusa group" but also M. neritoides and "M. perplexa group".  Moreover, they documented a contrasting ecological difference in their 2021 paper: "We found that M. labio was dominant in sheltered habitats and M. perplexa was dominant in wave-exposed habitats, while M. confusa showed no habitat specificity. This indicates that M. labio and M. perplexa are habitat specialists regarding wave exposure, while M. confusa is a generalist." In the previous review, it was pointed out that other sequences in GenBank are assigned to M. confusa, a species that is accepted in WoRMS. The authors cannot simply assert that M. labio is one species because Zhao et al. (2017) said so. When there is a controversy, the authors need to address this information near the beginning of their introduction to this taxon. I learned quite a bit in the responses from the authors that I wished was included in either version of the authors' manuscript. Now there are a couple of sentences near the end (!) of the manuscript: We incorporated sequences published by other studies into our analysis and found that they can be divided into two groups, lineage I and II. Among the individuals in the lineage II group, Zhao et al. [20] identified them as M. labio, while    Yamazaki et al. [20, 21] identified them as M. confusa. Due to the wide distribution range of this group, in our study, to avoid confusion in the classification system and to focus on the population's historical dynamics, we agree with the viewpoint of Zhao et al. [20] and consider the individuals in lineage II as M. labio (Figure S1, Table S2).

    Please, do not make a reader find out about that M. labio is being treated more broadly than is currently accepted in WoRMS. The authors still treat it as a single species for the entire manuscript despite apparent genetic evidence to the contrary that all three reviewers noticed. It seems that the case could be easily made for even more than two species. An appropriate place to bring up details about the accepted taxonomy in WoRMs, as earlier advocated by Yamazaki et al. (cited as 21), is in line 92, where the authors misrepresent their conclusion; they imply that Yamazaki and co-authors concluded there are two lineages within M. labio: " Another similar study proposed that M. labio was primarily separated into two lineages with a boundary between the Japanese mainland and the Ryukyu Islands [21]. The genetic structure of M. labio is also influenced by ecological factors such as habitat specificity and adaptability. Molecular studies have suggested lower genetic diversity and greater genetic differentiation in M. labio, possibly due to these ecological factors [22]."

    By not even explaining that there is a controversy that frankly seems to be favoring the two species hypothesis, judging from the acceptance of M. confusa in WoRMS, the authors appear to be overly stubborn to not explain and justify, in the Introduction, their decision to accept M. confusa as a synonym of M. labio, contrary to its treatment in WoRMS. By waiting until the last paragraph of their manuscript to bring up the current controversy in the literature, and never mentioning the accepted species ironically named, M. confusa, they have contributed to "confusion in the classification system." This is unacceptable.

    However, I would personally have no problem if the authors could, in a more substantial revision, sufficiently justify their choice of the broader species concept used by Zhao et al. (2017), not by assertion but based on evidence from their results. For example, they could further cite the morphometric conclusions in the later paper by Zhao et al. (2019), which at first glance to me seems to show substantial overlap in shell shape at least between the Zhao et al. (2017) "clades." There is also the issue that the analysis of Yamazaki et al. (2017) does not firmly support (bootstrap support is < 50%) the monophyly of their restricted M. labio, whereas, their M. confusa (with M. "australis" - Ogasawara) has 83% bootstrap support. From the haplotype network figure with redrawn Lineage "I" vs. "II" boundaries, it is possible that "I-C" sometimes groups with "I-B" + "I-A" and sometimes groups with "II" and this alternative topology needs to be considered. This analysis has added more geographic sampling that could be relevant to resolving these basal branching patterns of the "I" + "II" grouping, with or without the Ogasawara sequences included within "II."

Here are more technical suggestions of how a long-branch part of the ingroup taxon can "bounce around" in the tree and reduce resolution compared to if it is trimmed from the analysis. If combined phylogenetic estimates support the Ogasawara branch as part of the ingroup, it is still likely to be a long branch. The inclusion of a highly divergent ingroup sequence can obscure resolution of branching pattern, so I would recommend exploring deleting or restoring the Ogasawara samples and report on the impact on the apparent "I" vs. "II" basal split. Overall, the authors could find in a combined 16S+cox1 result that there is no clear evidence for splitting these lineages into only two species, instead supporting an unresolved polytomy of multiple lineages. The latter finding might support a broad M. labio species concept. But they need to include more proximal outgroups to evaluate this possibility. I recommend they include outgroups that arise from two nodes "below" the ingroup, which might or might not include M. sp. from the Ogasawara Islands as the most proximal outgroup. That is a result of the analysis, not something that is declared.

    I felt like I had to dig out important details on previous molecular systematics and biogeography research on Monodonta that I should have been introduced to early on, along with a generally good description of the fluctuating land barriers to dispersal during Pleistocene glacial intervals. The authors have made their lack of interest in the one- vs. two-species controversy clear, and I understand that there could be a case for a broad definition of a species. But in this case the species controversies happen to be highly relevant to their genetic study. The controversy cannot simply be ignored until the last paragraph. Some of the present results could be open to alternative interpretations if more than one species is represented in their samples, or at least the reader deserves to learn about them much earlier.

     As the authors point out, this is one of the most common snails on the shores of the study area. Previous authors, from studies dating to 2005 but most recently by Yamazaki et al. (2017; 2021), have at least convinced the knowledgeable editors at WoRMS (MolluscaBase) that both M. labio and M. confusa are currently accepted. Can the authors cite a study more recent than 2017 that has disputed the two-species interpretation? It is even more interesting that habitat differences exist between species or intraspecific lineages where they co-occur (Yamazaki et al. 2021). The possibility at least needs to be addressed as it relates to results of this study.

Response: We agree with the reviewers' suggestions. Based on the study by Yamazaki et al. (2021), we now consider the lineage II we initially identified in the article as M. confusa. As these two lineages represent different species, we have significantly modified our analysis in accordance with this interpretation. Consequently, we have conducted population genetic analyses separately for the two distinct species, each following its respective model.

Response:  As requested, we revised it. Following the reviewer's suggestions, we have significantly revised the section on species delineation and, as recommended, conducted separate analyses based on the concepts of the two species. This has been done to prevent readers from having the misconceptions mentioned above.

    With respect to the ABC software employed in this study, I am not directly familiar with the use of this particular software but I do think it must matter substantially if one chooses to group two species as if this was a priori a single group. It could easily inflate estimates such as effective population size if there is no gene flow between them. If there is previous evidence in the literature that substantial hybridization between M. labio and M. confusa, as accepted currently in WoRMS, then the situation could be different.

    It did not help that the authors have redundantly posted the same flawed assertion response of citing only a single study to justify their taxonomy to multiple objections raised in their response to reviewers. Because mitochondrial markers are maternally inherited with no opportunity to directly test for genetic connectivity, they cannot assume a potential for interbreeding across their "populations," and having Lineage "I" and "II" co-occurring in part of their study area might itself suggest that at least two species are present.

Response: We concur with the reviewers' comments that the two lineages (lineage I and II) correspond to Monodonta labio and M. confusa, respectively. We have reevaluated our population analysis accordingly and conducted ABC analyses separately for the two species to assess their historical population expansions.

    I have reached the conclusion that only another major revision would potentially make this an acceptable manuscript for publication. Their present revision is minimal and does not sufficiently address the criticisms made by all three reviewers. I am close to recommending the manuscript could be rejected if for no other reason that the authors seem think that historical cladogenetic events (i.e., speciation) are not relevant to their analysis of "populations within a single species" and "the processes of historical changes." Evolution is not only anagenesis (drift and selection), it also involves cladogenesis. When framed in a biogeographic context, this makes an important difference the authors need to address. On the other hand, I still think a more substantial manuscript revision than the authors have provided could be a valuable contribution to the literature on these common and interesting snails that have relatively brief planktonic larval duration.

Response: Thank you for the reviewers' suggestions. We appreciate your insights, and we have made significant revisions to the article based on your feedback. We have reanalyzed all the genetic data for the populations and have rewritten the entire paper accordingly.

Specific comments:

Line 50, is "repressively" the right word?

Response: As requested, we revised these sentences. “During major glaciations, when there is a sea level decline of approximately 120–140 m below the present level, the northwestern (NW) Pacific is thought to form a continuous land mass effectively closing the sea passage (the Korean Strait and Taiwan Strait), because of their geographical isolation, the East China Sea (ECS) and the South China Sea (SCS) are partially enclosed, with connections to the Pacific occurring through the Okinawa Trough and the Bashi Strait in a restricted manner.”

 Lines 72-79, I have no idea of the authors final text from, and at some other places where there are edits. Guessing at how fragments of old versions will be eliminated still leaves sentences that are convoluted and use the wrong words. I need to be able to read entire sentences. It is partly because of the confusing way this manuscript has been edited to leave in old text.

Response: In this revised version, we have removed the use of track changes in the main text to facilitate the reviewers' reading. The document with track changes will be provided as an attachment.

Line 85, "In contrast,…"

Response: As requested, we revised these sentences. “In contrast, a short planktonic stage in marine organisms has revealed a clear genetic structure, such as Japanese turban shell (Turbo (Batillus) cornutus) [17] and Moon Turban Snail (Lunella granulate) [18].”

Line 105, suggest "mainland China, Jeju Island in Korea [was missing], and representing the Ryukyu Islands and more northern localities within Japan."

Or: "extending comparisons from previous studies to both coasts of Taiwan, expanding sampling from mainland China, Jeju Island in Korea, the Ryukyu Islands and more northern localities within Japan." Also note that "Ryukyu" as a header in Table 1 needs to be fixed. Plus, it is part of Japan.

Response: As requested, we revised these sentences. “The mitochondrial COI + 16S gene, which was 1,340 bp long, was sequenced from 85 specimens of M. labio and 41 specimens of M. confusa collected from thirteen localities in Taiwan, mainland China, Jeju Island in Korea [was missing], and representing the Ryukyu Islands and more northern localities within Japan (Table 1 and Figure 1).”

Line 106, suggest "simulations, which offer…"

Response: As requested, we revised these sentences. “Relying on the former molecular results, our objective was to use ABC simulations, which offers a framework for testing competing hypotheses to determine the best model of population divergence and demographics that fits the patterns of key demo-graphic parameters, such as changes in effective population size through time.”

Line 168-169: The manuscript in general is insufficiently proofread.

Response: We revised the manuscript again.

I did not make further specific comments.

Comments on the Quality of English Language

See comments for lines 72-79. There are similar other problems.

Response: Regarding the English editing issue, we agree that if the article is accepted, we will perform another round of English editing before publication.
